# STABILITY AND SHARPER RISK BOUNDS WITH CONVERGENCE RATE $O(1/n^2)$

## ABSTRACT

The sharpest known high probability excess risk bounds are up to $O(1/n)$ for empirical risk minimization and projected gradient descent via algorithmic stability (Klochkov & Zhivotovskiy, 2021). In this paper, we show that high probability excess risk bounds of order up to $O(1/n^2)$ are possible. We discuss how high probability excess risk bounds reach $O(1/n^2)$ under strong convexity, smoothness and Lipschitz continuity assumptions for empirical risk minimization, projected gradient descent and stochastic gradient descent. Besides, to the best of our knowledge, our high probability results on the generalization gap measured by gradients for nonconvex problems are also the sharpest.

## 1 INTRODUCTION

Algorithmic stability is a fundamental concept in learning theory (Bousquet & Elisseeff, 2002), which can be traced back to the foundational works of Vapnik & Chervonenkis (1974) and has a deep connection with learnability (Rakhlin et al., 2005; Shalev-Shwartz et al., 2010; Shalev-Shwartz & Ben-David, 2014). Simply speaking, we say that an algorithm is stable if a change in a single example in the training dataset leads to only a minor change in the output model. Stability often provides theoretical upper bounds on generalization error. The study of the relationship between stability and generalization enables us to theoretically understand and better control the behavior of the algorithm.

While providing in-expectation generalization error bounds through stability arguments is relatively straightforward, deriving high probability bounds presents a more significant challenge. However, these high probability bounds are crucial for understanding the robustness of optimization algorithms, as highlighted by recent works (Feldman & Vondrak, 2019; Bousquet et al., 2020; Klochkov & Zhivotovskiy, 2021). In practical scenarios, where we often train models a limited number of times, high probability bounds offer more informative insights than their in-expectation counterparts. Therefore, this paper focuses on improving high probability risk bounds through an exploration of algorithmic stability.

Let us start with some standard notations. We have a set of independent and identically distributed observations $S = \{z_1, \ldots, z_n\}$ sampled from a probability measure $\rho$ defined on a sample space $\mathcal{Z} := \mathcal{X} \times \mathcal{Y}$. Based on the training set $S$, our goal is to build a model $h : \mathcal{X} \mapsto \mathcal{Y}$ for prediction, where the model is determined by parameter $\mathbf{w}$ from parameter space $\mathcal{W} \subset \mathbb{R}^d$. The performance of a model $\mathbf{w}$ on an example $z$ can be quantified by a loss function $f(\mathbf{w}; z)$, where $f : \mathcal{W} \times \mathcal{Z} \mapsto \mathbb{R}_+$. Then the population risk and the empirical risk of $\mathbf{w} \in \mathcal{W}$, respectively as

$$F(\mathbf{w}) := \mathbb{E}_z\left[f(\mathbf{w}; z)\right], \quad F_S(\mathbf{w}) := \frac{1}{n} \sum_{i=1}^{n} f(\mathbf{w}; z_i),$$

where $\mathbb{E}_z$ denotes the expectation w.r.t. $z$. Let $\mathbf{w}^* \in \arg\min_{\mathbf{w} \in \mathcal{W}} F(\mathbf{w})$ be the model with the minimal population risk in $\mathcal{W}$ and let $A(S)$ be the output of a (possibly randomized) algorithm $A$ on the dataset $S$. Traditional generalization analysis aims to bound the generalization error $F(A(S)) - F_S(A(S))$ w.r.t the algorithm $A$ and the dataset $S$. Based on the technique developed by Feldman & Vondrak (2018; 2019), Bousquet et al. (2020) provide the sharpest high probability bounds of $O(L/\sqrt{n})$, where the loss function $f(\cdot, \cdot)$ is bounded by $L$. No matter how stable the algorithm is,

the high probability generalization bound will not be smaller than $O\left(L/\sqrt{n}\right)$. This is sampling error term scaling as $O\left(1/\sqrt{n}\right)$ that controls the generalization error (Klochkov & Zhivotovskiy, 2021).

A frequently used alternative to generalization bounds, that can avoid the sampling error, are the excess risk bounds. The excess risk of algorithm $A$ with respect to the dataset $S$ is defined as $F(A(S)) - F(\mathbf{w}^*)$. This is crucial, as it can be decomposed into an optimization term and a generalization term (Klochkov & Zhivotovskiy, 2021). To establish a bound on the excess risk, it is necessary to take into account both the generalization error and the optimization error. Recently, Klochkov & Zhivotovskiy (2021) developed the results in Bousquet et al. (2020) and provided the best high probability excess risk bounds of order up to $O\left(\log n/n\right)$ for empirical risk minimization (ERM) and projected gradient descent (PGD) algorithms via algorithmic stability. Since then, numerous of papers, for example (Yuan & Li, 2023; 2024), extended their results to various settings within their method. However, all the excess risk upper bounds derived from their method can not be smaller than $O\left(\log n/n\right)$. Considering this, we would like to have a question:

*Can algorithmic stability provide high probability excess risk bounds with the rate beyond $O(1/n)$?*

The main results of this paper answer this question positively. To address this question, this paper technically provides a tighter generalization error bound of the gradients, based on which, we build the relationship between algorithmic stability and excess risk bounds. We also establish the first high probability excess risk bounds that are dimension-free with the rate $O\left(1/n^2\right)$ for ERM, PGD and stochastic gradient descent (SGD).

Our contributions can be summarized as follows:

- We provide sharper high probability upper bounds for the generalization gap between the population risk and the empirical risk of gradients in general nonconvex settings. Currently, there is only one work (Fan & Lei, 2024) that investigates the high-probability generalization error bounds of gradients in nonconvex problems using algorithmic stability, which establishes an upper bound of $O\left(M/\sqrt{n}\right)$, where $M$ denotes the maximum gradient of loss functions value. In contrast, this paper provides tighter results. Our coefficient before $1/\sqrt{n}$ depends on the variance of the gradients of the loss functions under the model optimized by algorithm $A$. As is well known, optimization algorithms typically yield parameters that approach the optimal solution, which significantly reduces the term compared to the maximum gradient.

- We build the relationship between algorithmic stability and excess risk bounds. Our results can provide a more granular analysis dependent on optimal parameters. Compared to existing work (Klochkov & Zhivotovskiy, 2021), we can achieve the excess risk bounds with the rate of $O\left(1/n^2\right)$ under specific conditions. Under the same algorithmic stability, our results also perform tighter. To the best of our knowledge, these are the first dimension-free results with the order $O\left(1/n^2\right)$ in high probability risk bounds via algorithmic stability.

- Using our method, we derive the first dimension-free high probability excess risk bounds of $O\left(1/n^2\right)$ for ERM, PGD, and SGD, addressing an open problem posed in Xu & Zeevi (2024). While they achieved $O\left(1/n^2\right)$ excess risk bounds on ERM and GD algorithms through uniform convergence, their approach requires that the sample size satisfies $n = \Omega(d)$. In contrast, we successfully obtain $O\left(1/n^2\right)$ bounds that do not depend on the dimensionality using algorithmic stability.

## 2 RELATED WORK

Algorithmic stability is a classical approach in generalization analysis, which can be traced back to the foundational works of (Vapnik & Chervonenkis, 1974). It gave the generalization bound by analyzing the sensitivity of a particular learning algorithm when changing one data point in the dataset. Modern method of stability analysis was established by Bousquet & Elisseeff (2002), where they presented an important concept called uniform stability. Since then, a lot of works based on uniform stability have emerged. On one hand, generalization bounds with algorithmic stability have been significantly improved by Feldman & Vondrak (2018; 2019); Bousquet et al. (2020); Klochkov & Zhivotovskiy (2021). On the other hand, different algorithmic stability measures such as uniform argument stability (Liu et al., 2017; Bassily et al., 2020), uniform stability in gradients (Lei, 2023;

Fan & Lei, 2024), on average stability (Shalev-Shwartz et al., 2010; Kuzborskij & Lampert, 2018), hypothesis stability (Bousquet & Elisseeff, 2002; Charles & Papailiopoulos, 2018), hypothesis set stability (Foster et al., 2019), pointwise uniform stability (Fan & Lei, 2024), PAC-Bayesian stability (Li et al., 2020), locally elastic stability (Deng et al., 2021), and collective stability (London et al., 2016) have been developed. Most of them provided the connection on stability and generalization in expectation. Bousquet & Elisseeff (2002); Elisseeff et al. (2005); Feldman & Vondrak (2018; 2019); Bousquet et al. (2020); Klochkov & Zhivotovskiy (2021); Yuan & Li (2023; 2024); Fan & Lei (2024) considered high probability bounds. In this paper, we further develop the theory of stability and generalization by pushing the boundaries of stability analysis. We seek to determine how tight the stability performance bounds can be. To address this issue, we firstly provide stability bounds with the order $O\left(1/n^2\right)$.

## 3 STABILITY AND GENERALIZATION

In this section, we develop a novel concentration inequality which provides $p$-moment bound for sums of vector-valued functions. For a real-valued random variable $Y$, the $L_p$-norm of $Y$ is defined by $\|Y\|_p := (\mathbb{E}[|Y|^p])^{\frac{1}{p}}$. Similarly, let $\|\cdot\|$ denote the norm in a Hilbert space $\mathcal{H}$. For a random variable $X$ taking values in a Hilbert space, the $L_p$-norm of $X$ is defined by $\|\|\mathbf{X}\|\|_p := (\mathbb{E}[\|\mathbf{X}\|^p])^{\frac{1}{p}}$.

### 3.1 A MOMENT BOUND FOR SUMS OF VECTOR-VALUED FUNCTIONS

Here we present our sharper moment bound for sums of vector-valued functions of $n$ independent variables.

**Theorem 1.** *Let $\mathbf{Z} = (Z_1, \ldots, Z_n)$ be a vector of independent random variables each taking values in $\mathcal{Z}$, and let $\mathbf{g}_1, \ldots, \mathbf{g}_n$ be some functions: $\mathbf{g}_i : \mathcal{Z}^n \mapsto \mathcal{H}$ such that the following holds for any $i \in [n]$:*

- $\|\mathbb{E}[\mathbf{g}_i(\mathbf{Z})|Z_i]\| \leq G$ *a.s.*

- $\mathbb{E}\left[\mathbf{g}_i(\mathbf{Z})|Z_{[n]\setminus\{i\}}\right] = 0$ *a.s.,*

- $\mathbf{g}_i$ *satisfies the bounded difference property with $\beta$, namely, for any $i = 1, \ldots, n$, the following inequality holds*

$$\sup_{z_1,\ldots,z_n,z_j'} \|\mathbf{g}_i(z_1, \ldots, z_{j-1}, z_j, z_{j+1}, \ldots, z_n) - \mathbf{g}_i(z_1, \ldots, z_{j-1}, z_j', z_{j+1}, \ldots, z_n)\| \leq \beta. \quad (1)$$

*Then, for any $p \geq 2$, we have*

$$\left\|\left\|\sum_{i=1}^n \mathbf{g}_i\right\|\right\|_p \leq 2(\sqrt{2p} + 1)\sqrt{n}G + 4 \times 2^{\frac{1}{2p}}\left(\sqrt{\frac{p}{e}}\right)(\sqrt{2p} + 1)n\beta\lceil\log_2 n\rceil.$$

**Remark 1.** We start to compare with existing results. The proof is motivated by Bousquet et al. (2020); Klochkov & Zhivotovskiy (2021). Yuan & Li (2023; 2024) have also explored several related problems based on this approach. However, all of them focus specifically on upper bounds for sums of real-valued functions. The result most closely related to Theorem 1 is provided by Fan & Lei (2024). Under the same assumptions, Fan & Lei (2024) established the following inequality[1]

$$\left\|\left\|\sum_{i=1}^n \mathbf{g}_i\right\|\right\|_p \leq 2(\sqrt{2} + 1)\sqrt{np}G + 4(\sqrt{2} + 1)np\beta\lceil\log_2 n\rceil. \quad (2)$$

It is easy to verify that our result is tighter than result provided by Fan & Lei (2024) for both the first and second term. Comparing Theorem 1 with (2), the larger $p$ is, the tighter our result is relative to (2). In the worst case, when $p = 2$, the constant of our first term is $0.879$ times tighter than (2), and the constant of our second term is $0.634$ times tighter than (2). This is because we derive the optimal Marcinkiewicz-Zygmund's inequality for random variables taking values in a Hilbert space in the proof.

---

[1]They assume $n = 2^k, k \in \mathbb{N}$. Here we give the version of their result with general $n$.

Although Theorem 1 seems to the constant-level improvement, considering that this theorem has other broad applications, we report this result as well. On the other hand, the proof was challenging, as it involved establishing the best constant in Marcinkiewicz-Zygmund's inequality for random variables taking values in a Hilbert space, which has its foundations in Khintchine-Kahane's inequality. To prove the best constant, we utilized Stirling's formula for the Gamma function to construct appropriate functions, establishing both upper and lower bounds. Then using Mean Value Theorem, this approach ultimately led to the convergence of the constant as $p$ approaches infinity.

## 3.2 Sharper Generalization Bounds in Gradients

In this subsection, we come to the generalization bound in gradients. Let $\mathbf{w}^* \in \arg\min_{\mathbf{w} \in \mathcal{W}} F(\mathbf{w})$ be the model with the minimal population risk in $\mathcal{W}$ and $\mathbf{w}^*(S) \in \arg\min_{\mathbf{w} \in \mathcal{W}} F_S(\mathbf{w})$ be the model with the minimal empirical risk w.r.t. dataset $S$. Let $\|\cdot\|_2$ denote the Euclidean norm and $\nabla g(\mathbf{w})$ denote a subgradient of $g$ at $\mathbf{w}$. We denote $S = \{z_1, \ldots, z_n\}$ to be a set of independent random variables each taking values in $\mathcal{Z}$ and $S' = \{z_1', \ldots, z_n'\}$ be its independent copy. For any $i \in [n]$, define $S^{(i)} = \{z_i, \ldots, z_{i-1}, z_i', z_{i+1}, \ldots, z_n\}$ be a dataset replacing the $i$-th sample in $S$ with another i.i.d. sample $z_i'$.

We introduce some basic definitions here, and we want to emphasize that our main Theorem 2 and Theorem 3 do not need smoothness assumption and Polyak-Lojasiewicz condition, indicating their potential applications within the nonconvex problems as well.

**Definition 1.** *Let $f : \mathcal{W} \mapsto \mathbb{R}$. Let $M, \gamma, \mu > 0$.*

- *We say $f$ is $M$-Lipschitz if*

$$|f(\mathbf{w}) - f(\mathbf{w}')| \leq M\|\mathbf{w} - \mathbf{w}'\|, \forall \mathbf{w}, \mathbf{w}' \in \mathcal{W}.$$

- *We say $f$ is $\gamma$-smooth if*

$$\|\nabla f(\mathbf{w}) - \nabla f(\mathbf{w}')\|_2 \leq \gamma\|\mathbf{w} - \mathbf{w}'\|_2, \quad \forall \mathbf{w}, \mathbf{w}' \in \mathcal{W}.$$

- *Let $f^* = \min_{\mathbf{w} \in \mathcal{W}} f(\mathbf{w})$. We say $f$ satisfies the Polyak-Lojasiewicz (PL) condition with parameter $\mu > 0$ on $\mathcal{W}$ if*

$$f(\mathbf{w}) - f^* \leq \frac{1}{2\mu}\|\nabla f(\mathbf{w})\|_2^2, \quad \forall \mathbf{w} \in \mathcal{W}.$$

Then we define uniform stability in gradients.

**Definition 2** (Uniform Stability in Gradients)**.** *Let $A$ be a randomized algorithm. We say $A$ is $\beta$-uniformly-stable in gradients if for all neighboring datasets $S, S^{(i)}$, we have*

$$\sup_z \left\|\nabla f(A(S); z) - \nabla f(A(S^{(i)}); z)\right\|_2 \leq \beta. \tag{3}$$

**Remark 2.** Gradient-based stability was firstly introduced by Lei (2023); Fan & Lei (2024) to describe the generalization performance for nonconvex problems. In nonconvex problems, we can only find a local minimizer by optimization algorithms which may be far away from the global minimizer. Instead, the convergence of $\|\nabla F_S(A(S))\|_2$ was often studied in the optimization community (Ghadimi & Lan, 2013; Foster et al., 2018). Since the population risk of gradients $\|\nabla F(A(S))\|_2$ can be decomposed as the convergence of $\|\nabla F_S(A(S))\|_2$ and the generalization gap $\|\nabla F(A(S)) - \nabla F_S(A(S))\|_2$, the generalization analysis of $\|\nabla F(A(S)) - \nabla F_S(A(S))\|_2$ is not only useful in the derivations presented in this paper, but it is also crucial in nonconvex problems. This generalization gap can be achieved by uniform stability in gradients.

**Theorem 2** (Generalization via Stability in Gradients)**.** *Assume for any $z$, $f(\cdot, z)$ is $M$-Lipschitz. If $A$ is $\beta$-uniformly-stable in gradients, then for any $\delta \in (0, 1)$, the following inequality holds with probability at least $1 - \delta$*

$$\|\nabla F(A(S)) - \nabla F_S(A(S))\|_2$$

$$\leq 2\beta + \frac{4M\left(1 + e\sqrt{2\log(e/\delta)}\right)}{\sqrt{n}} + 8 \times 2^{\frac{1}{4}}(\sqrt{2} + 1)\sqrt{e}\beta \lceil \log_2 n \rceil \log(e/\delta).$$

**Remark 3.** Theorem 2 is a direct application via Theorem 1 where we denote the vector functions $\mathbf{g}_i(S) = \mathbb{E}_{z_i'} \left[ \mathbb{E}_Z \left[ \nabla f(A(S^{(i)}), Z) \right] - \nabla f(A(S^{(i)}), z_i) \right]$ and find that $\mathbf{g}_i(S)$ satisfies all the assumptions in Theorem 1. As a comparison, (Fan & Lei, 2024, Theorem 3) also developed high probability bounds under same assumptions, but our bounds are sharper since our moment inequality for sums of vector-valued functions are tighter as we have discussed in Remark 1. However, both Theorem 2 and Fan & Lei (2024) do not address the issue of the coefficient $O\left(1/\sqrt{n}\right)$ being dependent on $M$, which leads to a relationship with the maximum value of the gradients. Next, we derive sharper generalization bound of gradients under same assumptions.

**Theorem 3** (Sharper Generalization via Stability in Gradients). *Assume for any $z$, $f(\cdot, z)$ is $M$-Lipschitz. If $A$ is $\beta$-uniformly-stable in gradients, then for any $\delta \in (0,1)$, the following inequality holds with probability at least $1 - \delta$*

$$\|\nabla F(A(S)) - \nabla F_S(A(S))\|_2$$
$$\leq \sqrt{\frac{4\mathbb{E}_Z \left[\|\nabla f(A(S); Z)\|_2^2\right] \log \frac{6}{\delta}}{n}} + \sqrt{\frac{\left(\frac{1}{2}\beta^2 + 32n\beta^2 \log(3/\delta)\right) \log \frac{6}{\delta}}{n}} + \frac{M \log \frac{6}{\delta}}{n}$$
$$+ 16 \times 2^{\frac{3}{4}} \sqrt{e}\beta \left\lceil \log_2 n \right\rceil \log\left(3e/\delta\right) + 32\sqrt{e}\beta \left\lceil \log_2 n \right\rceil \sqrt{\log 3e/\delta}.$$

**Remark 4.** We begin by comparing our work with existing work. Note that the factor in both Theorem 2 and (Fan & Lei, 2024, Theorem 3) before $1/\sqrt{n}$ is $O\left(M\sqrt{\log\left(e/\delta\right)}\right)$, which depends on the max value of gradients $M$. However, under the same assumptions, the factor in Theorem 3 before $1/\sqrt{n}$ is $O\left(\sqrt{\mathbb{E}_Z \left[\|\nabla f(A(S); Z)\|_2^2\right] \log 1/\delta} + \beta \log(1/\delta)\right)$, not involving the possibly large term $M$. $\mathbb{E}_Z \left[\|\nabla f(A(S); Z)\|_2\right]$ can be interpreted as the variance of the gradients of the loss functions under the model optimized by algorithm $A$. As is known, optimization algorithms often provide parameters approaching the optimal solution, which make the term $\mathbb{E}_Z[\|\nabla f(A(S); Z)\|_2^2]$ much more smaller than $M$.

We want to emphasize that Theorem 3 addresses the relationship between algorithmic stability and generalization bounds in nonconvex settings, providing a bound that depends on the variance of the population risk of gradients rather than the maximum gradient $M$. Although our primary focus is not on the latter aspect of nonconvex settings. In fact, exploring high-probability stability for specific algorithms in nonconvex setting is indeed challenging. Current methods, such as those based on Bassily et al. (2020); Lei (2023), show that random algorithms under nonconvex smooth assumptions are $O(T/n)$-stable. However, this bound becomes less meaningful when the number of iterations exceeds $n$. We believe that further investigation into algorithm stability in nonconvex scenarios is valuable and worthy of exploration.

Here we give the proof sketch of Theorem 3, which is motivated by the analysis in Klochkov & Zhivotovskiy (2021). The key idea is to build vector functions $\mathbf{q}_i(S) = \mathbf{h}_i(S) - \mathbb{E}_{S\{z_i\}}[\mathbf{h}_i(S)]$ where we define vector functions $\mathbf{h}_i(S) = \mathbb{E}_{z_i'} \left[ \mathbb{E}_Z \left[ \nabla f(A(S^{(i)}), Z) \right] - \nabla f(A(S^{(i)}), z_i) \right]$. These functions satisfy all the assumptions in Theorem 1 and ensure the factor $M$ in Theorem 1 to $0$. Then the term $O(1/\sqrt{n})$ can be eliminated. Note that Eulidean norm of a vector being equal to $0$ does not necessarily imply that the vector itself is $0$. The distinction between them added complexity to the proof. Our proof required constructing vector functions to satisfy all assumptions in Theorem 1. Moreover, ensuring the factor $M$ in Theorem 1 approaches $0$ added a unique challenge. Utilizing the self-bounded property for vector functions also needs to consider the difference between vector's Eulidean norm being $0$ and the vector itself being $0$.

Besides, we introduce strong growth condition (SGC) here solely to clarify that Theorem 3 is tighter compared to Theorem 2. We only suppose this condition holds in Proposition 1.

**Definition 3** (Strong Growth Condition). *We say SGC holds if*

$$\mathbb{E}_Z \left[\|\nabla f(\mathbf{w}; Z)\|_2^2\right] \leq \rho \|\nabla F(\mathbf{w})\|_2^2.$$

**Remark 5.** There has been some related work that takes SGC into assumption Solodov (1998); Vaswani et al. (2019); Lei (2023). Vaswani et al. (2019) has proved that the squared-hinge loss with linearly separable data and finite support features satisfies the SGC.

**Proposition 1** (SGC case). *Let assumptions in Theorem 3 hold and suppose SGC holds. Then for any $\delta > 0$, with probability at least $1 - \delta$, we have*

$$\|\nabla F(A(S))\| \lesssim (1+\eta)\|\nabla F_S(A(S))\| + \frac{1+\eta}{\eta}\left(\frac{M}{n}\log\frac{1}{\delta} + \beta\log n \log\frac{1}{\delta}\right).$$

**Remark 6.** Proposition 1 build a connection between the population gradient error and the empirical gradient error under Lipschitz, nonconvex, nonsmooth and SGC case. When the algorithm is stable enough which means that $\beta = O(1/n)$, and performs well, the empirical risk of the gradient $\|\nabla F_S(A(S))\|_2$ can be zero. This implies that the population risk of gradients $\|\nabla F_S(A(S))\|_2$ can achieve $O(1/n)$. This result from Proposition 1 also helps to understand why Theorem 3 provides a better bound compared to Theorem 2. Specifically, using the inequality $\sqrt{ab} \leq \eta a + \frac{1}{\eta}b$ in the context of Theorem 3 allows us to derive Proposition 1. On the other hand, Theorem 2, combined with SGC and the assumption $\|\nabla F_S(A(S))\|_2 = 0, \beta = O(1/n)$, can only achieve a bound of $O(1/\sqrt{n})$ at best.

**Remark 7.** Finally, we claim a significant improvement over the results using uniform convergence, addressing an open problem posed by Xu & Zeevi (2024), namely achieving a bound of the same order that is independent of the dimension $d$. Uniform convergence is another popular tool in statistical learning theory for generalization analysis and best high probability bounds (Xu & Zeevi, 2024) based on uniform convergence is

$$\|\nabla F(A(S)) - \nabla F_S(A(S))\|_2$$
$$\lesssim \sqrt{\frac{\mathbb{E}_Z\left[\nabla\|f(\mathbf{w}^*;Z)\|_2^2\right]\log(1/\delta)}{n}} + \frac{\log(1/\delta)}{n} + \max\left\{\|A(S) - \mathbf{w}^*\|_2, \frac{1}{n}\right\}\left(\sqrt{\frac{d}{n}} + \frac{d}{n}\right), \quad (4)$$

which is the optimal result when we only consider the order of $n$. Uniform convergence results are related to the dimension $d$, which are unacceptable in high-dimensional learning problems. Note that (4) requires an additional smoothness-type assumption. As a comparison, when $f$ is $\gamma$-smoothness, our result in Theorem 3 can be easily derived as

$$\|\nabla F(A(S)) - \nabla F_S(A(S))\|_2$$
$$\lesssim \beta\log n\log(1/\delta) + \frac{\log(1/\delta)}{n} + \sqrt{\frac{\mathbb{E}_Z\left[\nabla\|f(\mathbf{w}^*;Z)\|_2^2\right]\log(1/\delta)}{n}} + \|A(S) - \mathbf{w}^*\|\sqrt{\frac{\log(1/\delta)}{n}}.$$

Above inequality also holds in nonconvex problems and implies that when the uniformly stable in gradients parameter $\beta$ is smaller than $1/\sqrt{n}$, our bound is tighter than (4) and is dimension independent.

### 3.3 Sharper Excess Risk Bounds

In this subsection, we proceed to introduce the PL condition, deriving the sharper excess risk bounds.

**Theorem 4.** *Let assumptions in Theorem 3 hold. Suppose the function $f$ is $\gamma$-smooth and the population risk $F$ satisfies the PL condition with parameter $\mu$. $\mathbf{w}^*$ is the projection of $A(S)$ onto the solution set $\arg\min_{\mathbf{w}\in\mathcal{W}} F(\mathbf{w})$. Then for any $\delta \in (0,1)$, when $n \geq \frac{16\gamma^2\log\frac{6}{\delta}}{\mu^2}$, with probability at least $1 - \delta$, we have*

$$F(A(S)) - F(\mathbf{w}^*) \lesssim \|\nabla F_S(A(S))\|_2^2 + \frac{F(\mathbf{w}^*)\log(1/\delta)}{n} + \frac{\log^2(1/\delta)}{n^2} + \beta^2\log^2 n\log^2(1/\delta).$$

**Remark 8.** Before explaining Theorem 4, I firstly introduce that the analysis of stability generalization consists of two parts: (a) the relationship between algorithmic stability and risk bounds, and (b) the stability of a specific algorithm. When analyzing the stability of particular algorithms, we often involve optimization analysis, especially when considering excess risk bounds. Theorem 4 focuses on the first part.

Theorem 4 implies that excess risk can be bound by the optimization gradient error $\|\nabla F_S(A(S))\|_2$ and uniform stability in gradients $\beta$. Note that the assumption $F(\mathbf{w}^*) = O(1/n)$ is common and can be found in Srebro et al. (2010); Lei & Ying (2020); Liu et al. (2018); Zhang et al. (2017); Zhang & Zhou (2019). This is natural since $F(\mathbf{w}^*)$ is the minimal population risk. On the other hand,

we can derive that under $\mu$-strongly convex and $\gamma$-smooth assumptions for the objective function $f$, uniform stability in gradients can be bounded of order $O(1/n)$ for ERM and PGD. Thus high probability excess risk can be bounded of order up to $O\left(1/n^2\right)$ under these common assumptions via algorithmic stability.

Comparing with current best related work (Klochkov & Zhivotovskiy, 2021), they only need the assumption of bounded loss function for the relationship between algorithmic stability and risk bounds. However, Our results provide a more granular analysis dependent on optimal parameters. On one hand, when the algorithm's stability $\beta = O(1/\sqrt{n})$ the upper bound, according to Klochkov & Zhivotovskiy (2021), can at most reach the order of $1/\sqrt{n}$ due to the algorithm's stability constraints. In contrast, our result shows that even under the assumption that $F(\mathbf{w}^*) = O(1)$, treating $F(\mathbf{w}^*)$ as a constant, we can achieve an order of $O(1/n)$. On the other hand, their result is insensitive to the stability parameter being smaller than $O(1/n)$ and their best rates can only up to $O(1/n)$. Our results can be up to $O(1/n^2)$ under some specific assumptions. We will discuss in Section 4.

Although we involve extra smoothness and PL condition assumptions, these assumptions are also common in optimization community and analyzing the stability of algorithms. For example, Klochkov & Zhivotovskiy (2021) introduced assumptions of strong convexity and smoothness in their study of the stability and optimization results of the PGD algorithm. However, their method did not fully leverage these assumptions when establishing the relationship between algorithmic stability and risk bounds. This is the fundamental reason why our results outperform theirs without introducing additional assumptions. Our work can fully utilize these assumptions.

## 4 APPLICATION

In this section, we analysis stochastic convex optimization with strongly convex losses. The most common setting is where at each round, the learner gets information on $f$ through a stochastic gradient oracle (Rakhlin et al., 2012). To derive uniform stability in gradients for algorithms, we firstly introduce the strongly convex assumption.

**Definition 4.** *We say $f$ is $\mu$-strongly convex if*

$$f(\mathbf{w}) \geq f(\mathbf{w}') + \langle \mathbf{w} - \mathbf{w}', \nabla f(\mathbf{w}') \rangle + \frac{\mu}{2} \|\mathbf{w} - \mathbf{w}'\|_2^2, \quad \forall \mathbf{w}, \mathbf{w}' \in \mathcal{W}.$$

### 4.1 EMPIRICAL RISK MINIMIZER

Empirical risk minimizer is one of the classical approaches for solving stochastic optimization (also referred to as sample average approximation (SAA)) in machine learning community. The following lemma shows the uniform stability in gradient for ERM under $\mu$-strongly convexity and $\gamma$-smoothness assumptions.

**Lemma 1** (Stability of ERM). *Suppose the objective function $f$ is $M$-Lipschitz, $\mu$-strongly-convex and $\gamma$-smooth. Let $\hat{\mathbf{w}}^*(S^{(i)})$ be the ERM of $F_{S^{(i)}}(\mathbf{w})$ that denotes the empirical risk on the samples $S^{(i)} = \{z_1, ..., z_i', ..., z_n\}$ and $\hat{\mathbf{w}}^*(S)$ be the ERM of $F_S(\mathbf{w})$ on the samples $S = \{z_1, ..., z_i, ..., z_n\}$. For any $S^{(i)}$ and $S$, there holds the following uniform stability bound of ERM:*

$$\forall z \in \mathcal{Z}, \quad \left\| \nabla f(\hat{\mathbf{w}}^*(S^{(i)}); z) - \nabla f(\hat{\mathbf{w}}^*(S); z) \right\|_2 \leq \frac{4M\gamma}{n\mu}.$$

Then, we present the application of our main sharper Theorem 3. In the strongly convex and smooth case, we provide a up to $O\left(1/n^2\right)$ high probability excess risk guarantee valid for any algorithms depending on the optimal population error $F(\mathbf{w}^*)$.

**Theorem 5.** *Let assumptions in Theorem 4 and Lemma 1 hold. Suppose the function $f$ is nonnegative. Then for any $\delta \in (0, 1)$, when $n \geq \frac{16\gamma^2 \log \frac{6}{\delta}}{\mu^2}$, with probability at least $1 - \delta$, we have*

$$F(\hat{\mathbf{w}}^*(S)) - F(\mathbf{w}^*) \lesssim \frac{F(\mathbf{w}^*) \log (1/\delta)}{n} + \frac{\log^2 n \log^2(1/\delta)}{n^2}.$$

**Remark 9.** Theorem 5 shows that when the objective function $f$ is $\mu$-strongly convex, $\gamma$-smooth and nonnegative, high probability risk bounds can even up to $O\left(1/n^2\right)$ for ERM. The most related work to ours is Zhang et al. (2017). They also obtain the $O\left(1/n^2\right)$-type bounds for ERM by

uniform convergence of gradients approach under the same assumptions. However, they need the sample number $n = \Omega(\gamma d/\mu)$, which is related to the dimension $d$. Our risk bounds are dimension independent and only require the sample number $n = \Omega(\gamma^2/\mu^2)$. Comparing with Klochkov & Zhivotovskiy (2021), we add two assumptions, smoothness and $F(\mathbf{w}^*) = O(1/n)$, the later of which is a common assumption towards sharper risk bounds (Srebro et al., 2010; Lei & Ying, 2020; Liu et al., 2018; Zhang et al., 2017; Zhang & Zhou, 2019), but our bounds are also tighter, from $O(1/n)$ to $O\left(1/n^2\right)$.

Our results are asymptotically optimal, which aligns with existing theories. According to the classical asymptotic theory, under some local regularity assumptions, when $n \to \infty$, it is shown in the asymptotic statistics monographs Van der Vaart (2000) that

$$\sqrt{n}(\hat{\mathbf{w}}^*(S) - \mathbf{w}^*) \xrightarrow{\rho} \mathcal{N}(0, \mathbf{H}^{-1}\mathbf{Q}\mathbf{H}^{-1}), \tag{5}$$

where $\hat{\mathbf{w}}^*(S)$ denotes the ERM algorithm, $\mathbf{H} = \nabla^2 F(\mathbf{w}^*)$, $\mathbf{Q}$ is the covariance matrix of the loss gradient at $\mathbf{w}^*$ (also called Fisher's information matrix): $\mathbf{Q} = \mathbb{E}[\nabla f(\mathbf{w}^*; z)\nabla f(\mathbf{w}^*; z)^T]$ ($\mathbf{A}^T$ denotes the transpose of a matrix $\mathbf{A}$), and $\xrightarrow{\rho}$ means convergence in distribution. The second-order Taylor expansion of the population risk around $\mathbf{w}^*$ then allows to derive the same asymptotic law for the scaled excess risk $2n(F(\hat{\mathbf{w}}^*(S)) - F(\mathbf{w}^*))$. Under suitable conditions, this asymptotic rate is usually theoretically optimal van der Vaart (1989). For example, when $f(\mathbf{w}; z)$ is a negative log-likelihood, this asymptotic rate matches the Hajek-LeCam asymptotic minimax lower bound Hájek (1972); Le Cam et al. (1972). We then analysis the result in Theorem 5. In the proof of Theorem 4, before we use the self-bounded smoothness property $\|\nabla f(\mathbf{w}^*; z)\|^2 \le 4\gamma f(\mathbf{w}^*; z)$, we get the following result for Theorem 5

$$F(\hat{\mathbf{w}}^*(S)) - F(\mathbf{w}^*) \lesssim \frac{\mathbb{E}[\|\nabla f(\mathbf{w}^*; z)\|^2]\log(1/\delta)}{\mu n} + \frac{\log^2(1/\delta)}{n^2}.$$

Our result is the finite sample version of the asymptotic rate (5), which characterizes the critical sample size sufficient to enter this "asymptotic regime". This is because the excess risk error $F(\hat{\mathbf{w}}^*(S)) - F(\mathbf{w}^*)$ can be approximated by the quadratic form $(\hat{\mathbf{w}}^*(S) - \mathbf{w}^*)^T \mathbf{H}(\hat{\mathbf{w}}^*(S) - \mathbf{w}^*)$. $1/\mu$ is a natural proxy for the inverse Hessian $\mathbf{H}^{-1}$, and $\mathbb{E}[\|\nabla f(\mathbf{w}^*; z)\|^2]$ is a natural proxy for Fisher's information matrix $\mathbf{Q}$. Furthermore, when discussing sample complexity, Xu & Zeevi (2024) constructed a simple linear model to demonstrate the constant-level optimality of the sample complexity lower bound $\Omega(d\beta^2/\mu^2)$ under such conditions. Our theorem further reveals, through the use of stability methods, that this complexity lower bound can be independent of the dimensionality $d$.

### 4.2 PROJECTED GRADIENT DESCENT

Note that when the objective function $f$ is strongly convex and smooth, the optimization error can be ignored. However, the generalization analysis method proposed by Klochkov & Zhivotovskiy (2021) does not use smoothness assumption, which only derive high probability excess risk bound of order $O(1/n)$ after $T = O(\log n)$ steps under strongly convex and smooth assumptions. In this subsection, we provide sharper risk bound under the same iteration steps, which is because our generalization analysis also fully utilized the smooth assumptions. Here we introduce the procedure of the PGD algorithm.

Let $\mathbf{w}_1 \in \mathbb{R}^d$ be an initial point and $\{\eta_t\}_t$ be a sequence of positive step sizes. PGD updates parameters by

$$\mathbf{w}_{t+1} = \Pi_{\mathcal{W}}\left(\mathbf{w}_t - \eta_t \nabla F_S\left(\mathbf{w}_t\right)\right),$$

where $\nabla F_S(\mathbf{w}_t)$ denotes a subgradient of $F$ w.r.t. $\mathbf{w}_t$ and $\Pi_{\mathcal{W}}$ is the projection operator onto $\mathcal{W}$.

**Lemma 2** (Stability of Gradient Descent). *Suppose the objective function $f$ is $M$-Lipschitz, $\mu$-strongly-convex and $\gamma$-smooth. Let $\mathbf{w}_t'$ be the output of $F_{S^{(i)}}(\mathbf{w})$ on $t$-th iteration on the samples $S^{(i)} = \{z_1, ..., z_i', ..., z_n\}$ in running PGD, and $\mathbf{w}_t$ be the output of $F_S(\mathbf{w})$ on $t$-th iteration on the samples $S = \{z_1, ..., z_i, ..., z_n\}$ in running PGD. Let the constant step size $\eta_t = 1/\gamma$. For any $S^{(i)}$ and $S$, there holds the following uniform stability bound of PGD:*

$$\forall z \in \mathcal{Z}, \quad \left\|\nabla f(\mathbf{w}_t^i; z) - \nabla f(\mathbf{w}_t; z)\right\|_2 \le \frac{2M\gamma}{n\mu}.$$

**Remark 10.** The derivations of Feldman & Vondrak (2019) in Section 4.1.2 (See also Hardt et al. (2016) in Section 3.4) imply that if the objective function $f$ is $\gamma$-smooth in addition to $\mu$-strongly convexity and $M$-Lipschitz property, then PGD with the constant step size $\eta = 1/\gamma$ is $\left(\frac{2M}{n\mu}\right)$-uniformly argument stable for any number of steps, which means that PGD is $\left(\frac{2M\gamma}{n\mu}\right)$-uniformly-stable in gradients regardless of iteration steps.

**Theorem 6.** *Let assumptions in Theorem 4 and Lemma 1 hold. Suppose the function $f$ is nonnegative. Let $\{\mathbf{w}_t\}_t$ be the sequence produced by PGD with $\eta_t = 1/\gamma$. Then for any $\delta \in (0, 1)$, when $n \geq \frac{16\gamma^2 \log \frac{6}{\delta}}{\mu^2}$, with probability at least $1 - \delta$, we have*

$$F(\mathbf{w}_{T+1}) - F(\mathbf{w}^*) \lesssim \left(1 - \frac{\mu}{\gamma}\right)^{2T} + \frac{F(\mathbf{w}^*)\log(1/\delta)}{n} + \frac{\log^2 n \log^2(1/\delta)}{n^2}.$$

*Furthermore, assume $F(\mathbf{w}^*) = O(\frac{1}{n})$ and let $T \asymp \log n$, we have*

$$F(\mathbf{w}_{T+1}) - F(\mathbf{w}^*) \lesssim \frac{\log^2 n \log^2(1/\delta)}{n^2}.$$

**Remark 11.** Theorem 6 shows that under the same assumptions as Klochkov & Zhivotovskiy (2021), our bound is $O\left(\frac{F(\mathbf{w}^*)\log(1/\delta)}{n} + \frac{\log^2 n \log^2(1/\delta)}{n^2}\right)$. Comparing with their bound $O\left(\frac{\log n \log(1/\delta)}{n}\right)$, we are sharper because $F(\mathbf{w}^*)$ is the minimal population risk, which is a common assumption towards sharper risk bounds (Srebro et al., 2010; Lei & Ying, 2020; Liu et al., 2018; Zhang et al., 2017; Zhang & Zhou, 2019). We use this assumption to demonstrate that under low-noise conditions, our bounds can achieve the tightest possible rate of $O(1/n^2)$.

## 4.3 STOCHASTIC GRADIENT DESCENT

Stochastic gradient descent optimization algorithm has been widely used in machine learning due to its simplicity in implementation, low memory requirement and low computational complexity per iteration, as well as good practical behavior. We provide the excess risk bounds for SGD using our method in this subsection. Here we introduce the procedure of the standard SGD algorithm.

Let $\mathbf{w}_1 \in \mathbb{R}^d$ be an initial point and $\{\eta_t\}_t$ be a sequence of positive step sizes. SGD updates parameters by

$$\mathbf{w}_{t+1} = \Pi_{\mathcal{W}}\left(\mathbf{w}_t - \eta_t \nabla f\left(\mathbf{w}_t; z_{i_t}\right)\right),$$

where $\nabla f(\mathbf{w}_t; z_{i_t})$ denotes a subgradient of $f$ w.r.t. $\mathbf{w}_t$ and $i_t$ is independently drawn from the uniform distribution over $[n] := \{1, 2, \ldots, n\}$.

**Lemma 3** (Stability of SGD). *Suppose the objective function $f$ is $M$-Lipschitz, $\mu$-strongly-convex and $\gamma$-smooth. Let $\mathbf{w}_t^i$ be the output of $F_{S^{(i)}}(\mathbf{w})$ on $t$-th iteration on the samples $S^{(i)} = \{z_1, ..., z_i', ..., z_n\}$ in running PGD and and $\mathbf{w}_t$ be the output of $F_S(\mathbf{w})$ on $t$-th iteration on the samples $S = \{z_1, ..., z_i, ..., z_n\}$ in running SGD. For any $S^{(i)}$ and $S$, there holds the following uniform stability bound of SGD:*

$$\left\|\nabla f(\mathbf{w}_t; z) - \nabla f(\mathbf{w}_t^i; z)\right\|_2 \leq 2\gamma\sqrt{\frac{2\epsilon_{opt}(\mathbf{w}_t)}{\mu}} + \frac{4M\gamma}{n\mu}, \quad \forall z \in \mathcal{Z},$$

*where $\epsilon_{opt}(\mathbf{w}_t) = F_S(\mathbf{w}_t) - F_S(\hat{\mathbf{w}}^*(S))$ and $\hat{\mathbf{w}}^*(S)$ is the ERM of $F_S(\mathbf{w})$.*

Next, we introduce a necessary assumption in stochastic optimization theory.

**Assumption 1.** *Assume the existence of $\sigma > 0$ satisfying*

$$\mathbb{E}_{i_t}\left[\|\nabla f(\mathbf{w}_t; z_{i_t}) - \nabla F_S(\mathbf{w}_t)\|_2^2\right] \leq \sigma^2, \quad \forall t \in \mathbb{N}, \tag{6}$$

*where $\mathbb{E}_{i_t}$ denotes the expectation w.r.t. $i_t$.*

**Remark 12.** Assumption 1 is a standard assumption from the stochastic optimization theory (Nemirovski et al., 2009; Ghadimi & Lan, 2013; Ghadimi et al., 2016; Kuzborskij & Lampert, 2018; Zhou et al., 2018; Bottou et al., 2018; Lei & Tang, 2021), which essentially bounds the variance of the stochastic gradients for dataset $S$.

**Theorem 7.** *Let assumptions in Theorem 4 and Lemma 3 hold. Suppose Assumption 1 holds and the objective function $f$ is nonnegative. Let $\{\mathbf{w}_t\}_t$ be the sequence produced by SGD with $\eta_t = \eta_1 t^{-\theta}, \theta \in (0,1)$ and $\eta_1 < \frac{1}{2\gamma}$. Then for any $\delta \in (0,1)$, when $n \geq \frac{16\gamma^2 \log \frac{6}{\delta}}{\mu^2}$, with probability at least $1 - \delta$, we have*

$$
\left( \sum_{t=1}^{T} \eta_t \right)^{-1} \sum_{t=1}^{T} \eta_t \|\nabla F(\mathbf{w}_t)\|_2^2
$$
$$
= \begin{cases}
O\left( \frac{\log^2 n \log^3 (1/\delta)}{T^{-\theta}} \right) + O\left( \frac{\log^2 n \log^2 (1/\delta)}{n^2} + \frac{F(\mathbf{w}^*) \log^2 (1/\delta)}{n} \right), & \text{if } \theta < 1/2 \\
O\left( \frac{\log^2 n \log^3 (1/\delta)}{T^{-\frac{1}{2}}} \right) + O\left( \frac{\log^2 n \log^2 (1/\delta)}{n^2} + \frac{F(\mathbf{w}^*) \log^2 (1/\delta)}{n} \right), & \text{if } \theta = 1/2 \\
O\left( \frac{\log^2 n \log^3 (1/\delta)}{T^{\theta-1}} \right) + O\left( \frac{\log^2 n \log^2 (1/\delta)}{n^2} + \frac{F(\mathbf{w}^*) \log^2 (1/\delta)}{n} \right), & \text{if } \theta > 1/2.
\end{cases}
$$

**Remark 13.** When $\theta < 1/2$, we take $T \asymp n^{2/\theta}$. When $\theta = 1/2$, we take $T \asymp n^4$ and when $\theta > 1/2$, we set $T \asymp n^{2/(1-\theta)}$. Then according to Theorem 7, the population risk of gradient is bounded by $O\left( \frac{\log^2 n \log^3 (1/\delta)}{n^2} + \frac{F(\mathbf{w}^*) \log^2 (1/\delta)}{n} \right)$. When $F(\mathbf{w}^*) = O(1/n)$, we can reach the $O(1/n^2)$ bounds. If we only need the $O(1/n)$ results, we can loose the assumptions as follows. Assume $F(\mathbf{w}^*) = O(1)$ and when $\theta < 1/2$, we take $T \asymp n^{1/\theta}$. When $\theta = 1/2$, we take $T \asymp n^2$ and when $\theta > 1/2$, we set $T \asymp n^{1/(1-\theta)}$. To the best of our knowledge, both $O(1/n^2)$ and $O(1/n)$ bounds are the first high probability population gradient bound $\|\nabla F(\mathbf{w}_t)\|_2$ for SGD via algorithmic stability.

**Theorem 8.** *Let Assumptions in Theorem 3 and Lemma 3 hold. Suppose Assumption 1 holds and the function $f$ is nonnegative. Let $\{\mathbf{w}_t\}_t$ be the sequence produced by SGD with $\eta_t = \frac{2}{\mu(t+t_0)}$ such that $t_0 \geq \max\left\{ \frac{4\gamma}{\mu}, 1 \right\}$. Then for any $\delta > 0$, when $n \geq \frac{16\gamma^2 \log \frac{6}{\delta}}{\mu^2}$, with probability at least $1 - \delta$, we have*

$$
F(\mathbf{w}_{T+1}) - F(\mathbf{w}^*) = O\left( \frac{\lceil \log_2 n \rceil^2 \log T \log^5 (1/\delta)}{T} \right) + O\left( \frac{\lceil \log_2 n \rceil^2 \log^2 (1/\delta)}{n^2} + \frac{F(w^*) \log (1/\delta)}{n} \right).
$$

*Furthermore, assume $T \asymp n^2$ and $F(\mathbf{w}^*) = O(\frac{1}{n})$, we have*

$$
F(\mathbf{w}_{T+1}) - F(\mathbf{w}^*) = O\left( \frac{\log^4 n \log^5 (1/\delta)}{n^2} \right).
$$

**Remark 14.** Theorem 8 implies that high probability risk bounds for SGD optimization algorithm can be up to $O(1/n^2)$ and the rate is dimension-free in high-dimensional learning problems. We compare Theorem 8 with most related work. For algorithmic stability, high probability risk bounds in Fan & Lei (2024) is up to $O(1/n)$ when choosing optimal iterate number $T$ for SGD optimization algorithm. To the best of knowledge, we are faster than all the existing bounds. When $T \asymp n$ and $F(\mathbf{w}^*) = O(1)$, our bound is $O(1/n)$, which is also the sharpest hight probability bound under $T \asymp n$ iterations. For comparison, there are two results in stability analysis that are similar to ours. One is the $O(1/n)$ result when $T = n$ Lei & Ying (2020), but it pertains to the expected version and also needs $F(\mathbf{w}^*) = 0$. The high-probability version is significantly more challenging. Currently, the best result under the high-probability version is also $O(1/n)$ (Li & Liu, 2021), but Li & Liu (2021) requires $T = n^2$ iterations.

## 5 CONCLUSION

In this paper, we improve a $p$-moment concentration inequality for sums of vector-valued functions. By carefully constructing functions, we apply this moment concentration to derive sharper generalization bounds in gradients in nonconvex problems, which can further be used to obtain sharper high probability excess risk bounds for stable optimization algorithms. In application, we study three common algorithms: ERM, PGD, SGD. To the best of our knowledge, we provide the sharpest high probability dimension independent $O(1/n^2)$-type for these algorithms. Comparisons with existing work can be found in Table 1 in Appendix.

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

Table 1: Summary of high probability excess risk bounds. All conclusions herein assume Lipschitz continuity, and all SGD algorithms presuppose bounded variance of the gradients; therefore, these two assumptions are omitted in the table. Abbreviations: uniform convergence $\to$ UC, algorithmic stability $\to$ AS, strongly convex $\to$ SC, low noice $\to$ LN, Polyak-Lojasiewicz condition $\to$ PL.

| Reference | Algorithm | Method | Assumptions | Sample Size | Bounds |
|---|---|---|---|---|---|
| Zhang et al. (2017) | ERM | UC | Smooth, SC, LN | $\Omega\left(\frac{\gamma d}{\mu}\right)$ | $O\left(\frac{1}{n^2}\right)$ |
| Xu & Zeevi (2024) | ERM | UC | Smooth, PL, LN | $\Omega\left(\frac{\gamma^2 d}{\mu^2}\right)$ | $O\left(\frac{1}{n^2}\right)$ |
| | PGD | UC | Smooth, PL, LN | $\Omega\left(\frac{\gamma^2 d}{\mu^2}\right)$ | $O\left(\frac{1}{n^2}\right)$ |
| Li & Liu (2021) | SGD | UC | Smooth, PL, LN | $\Omega\left(\frac{\gamma^2 d}{\mu^2}\right)$ | $O\left(\frac{1}{n^2}\right)$ |
| | | AS | Smooth, SC | - | $O\left(\frac{1}{n}\right)$ |
| Klochkov & Zhivotovskiy (2021) | ERM | AS | SC | - | $O\left(\frac{1}{n}\right)$ |
| | PGD | AS | Smooth, SC | - | $O\left(\frac{1}{n}\right)$ |
| This work | ERM | AS | Smooth, SC, LN | $\Omega\left(\frac{\gamma^2}{\mu^2}\right)$ | $O\left(\frac{1}{n^2}\right)$ |
| | PGD | AS | Smooth, SC, LN | $\Omega\left(\frac{\gamma^2}{\mu^2}\right)$ | $O\left(\frac{1}{n^2}\right)$ |
| | SGD | AS | Smooth, SC, LN | $\Omega\left(\frac{\gamma^2}{\mu^2}\right)$ | $O\left(\frac{1}{n^2}\right)$ |

## A  ADDITIONAL DEFINITIONS AND LEMMATA

**Lemma 4** (Equivalence of tails and moments for random vectors (Bassily et al., 2020)). *Let $X$ be a random variable with*

$$\|X\|_p \leq \sqrt{p}a + pb$$

*for some $a, b \geq 0$ and for any $p \geq 2$. Then for any $\delta \in (0,1)$ we have, with probability at least $1 - \delta$,*

$$|X| \leq e\left(a\sqrt{\log\left(\frac{e}{\delta}\right)} + b\log\frac{e}{\delta}\right).$$

**Lemma 5** (Vector Bernstein's inequality (Pinelis, 1994; Smale & Zhou, 2007)). *Let $\{X_i\}_{i=1}^n$ be a sequence of i.i.d. random variables taking values in a real separable Hilbert space. Assume that $\mathbb{E}[X_i] = \mu$, $\mathbb{E}[\|X_i - \mu\|^2] = \sigma^2$, and $\|X_i\| \leq M$, $\forall 1 \leq i \leq n$, then for all $\delta \in (0,1)$, with probability at least $1 - \delta$ we have*

$$\left\|\frac{1}{n}\sum_{i=1}^n X_i - \mu\right\| \leq \sqrt{\frac{2\sigma^2 \log(\frac{2}{\delta})}{n}} + \frac{M\log\frac{2}{\delta}}{n}.$$

**Definition 5** (Weakly self-Bounded Function). *Assume that $a, b > 0$. A function $f : \mathcal{Z}^n \mapsto [0, +\infty)$ is said to be $(a, b)$-weakly self-bounded if there exist functions $f_i : \mathcal{Z}^{n-1} \mapsto [0, +\infty)$ that satisfies for all $Z^n \in \mathcal{Z}^n$,*

$$\sum_{i=1}^n (f_i(Z^n) - f(Z^n))^2 \leq af(Z^n) + b.$$

**Lemma 6** ((Klochkov & Zhivotovskiy, 2021)). *Suppose that $z_1, \ldots, z_n$ are independent random variables and the function $f : \mathcal{Z}^n \mapsto [0, +\infty)$ is $(a, b)$-weakly self-bounded and the corresponding function $f_i$ satisfy $f_i(Z^n) \geq f(Z^n)$ for $\forall i \in [n]$ and any $Z^n \in \mathcal{Z}^n$. Then, for any $t > 0$,*

$$Pr(\mathbb{E}f(z_1, \ldots, z_n) \geq f(z_1, \ldots, z_n) + t) \leq \exp\left(-\frac{t^2}{2a\mathbb{E}f(z_1, \ldots, z_n) + 2b}\right).$$

**Definition 6.** *A Rademacher random variable is a Bernoulli variable that takes values $\pm 1$ with probability $\frac{1}{2}$ each.*

## B  SUMMARY OF OUR HIGH PROBABILITY EXCESS RISK BOUNDS.

Our high probability excess risk bounds can be summarized in Table 1.

## C  PROOFS OF SECTION 3

### C.1  PROOFS OF SUBSECTION 3.1

The proof of Theorem 1 is motivated by Bousquet et al. (2020), which need the Marcinkiewicz-Zygmund's inequality for random variables taking values in a Hilbert space and the McDiarmid's inequality for vector-valued functions.

Firstly, we derive the optimal constants in the Marcinkiewicz-Zygmund's inequality for random variables taking values in a Hilbert space.

**Lemma 7** (Marcinkiewicz-Zygmund's Inequality for Random Variables Taking Values in a Hilbert Space). *Let $\mathbf{X}_1, \ldots, \mathbf{X}_n$ be random variables taking values in a Hilbert space with $\mathbb{E}[\mathbf{X}_i] = 0$ for all $i \in [n]$. Then for $p \geq 2$ we have*

$$\left\| \left\| \sum_{i=1}^{n} \mathbf{X}_i \right\| \right\|_p \leq 2 \cdot 2^{\frac{1}{2p}} \sqrt{\frac{np}{e}} \left( \frac{1}{n} \sum_{i=1}^{n} \left\| \|\mathbf{X}_i\| \right\|_p^p \right)^{\frac{1}{p}}.$$

**Remark 15.** Comparing with Marcinkiewicz-Zygmund's inequality given by Fan & Lei (2024), we provide best constants. Next, we give the proof of Lemma 7.

The Marcinkiewicz-Zygmund's inequality can be proved by using its connection to Khintchine-Kahane's inequality. Thus, we introduce the best constants in Khintchine-Kahane's inequality for random variables taking values from a Hilbert space here.

**Lemma 8** (Best constants in Khintchine-Kahane's inequality in Hilbert space (Latała & Oleszkiewicz, 1994; Luo & Zhang, 2020)). *For all $p \in [2, \infty)$ and for all choices of Hilbert space $\mathcal{H}$, finite sets of vectors $\mathbf{X}_i, \ldots, \mathbf{X}_n \in \mathcal{X} \in \mathcal{H}$, and independent Rademacher variables $r_1, \ldots, r_n$,*

$$\left[ \mathbb{E} \left\| \sum_{i=1}^{n} r_i \mathbf{X}_i \right\|^p \right]^{\frac{1}{p}} \leq C_p \cdot \left[ \sum_{i=1}^{n} \|\mathbf{X}_i\|^2 \right]^{\frac{1}{2}},$$

*where $C_p = 2^{\frac{1}{2}} \left\{ \frac{\Gamma\left(\frac{p+1}{2}\right)}{\sqrt{\pi}} \right\}^{\frac{1}{p}}$.*

*Proof of Lemma 7.* The symmetrization argument goes as follows: Let $(r_1, \ldots, r_n)$ be i.i.d. with $\mathbb{P}(r_i = 1) = \mathbb{P}(r_i = -1) = 1/2$ and besides such that $r_1, \ldots, r_n$ and $(\mathbf{X}_1, \ldots, \mathbf{X}_n)$ are independent. Then by independence and symmetry, according to Lemma 1.2.6 of De la Pena & Giné (2012), conditioning on $(\mathbf{X}_1, \ldots, \mathbf{X}_n)$ yields

$$\mathbb{E} \left[ \left\| \sum_{i=1}^{n} \mathbf{X}_i \right\|^p \right] = 2^p \mathbb{E} \left[ \left\| \sum_{i=1}^{n} r_i \mathbf{X}_i \right\|^p \right] \leq 2^p \mathbb{E} \left[ \mathbb{E} \left[ \left\| \sum_{i=1}^{n} r_i \mathbf{X}_i \right\|^p \middle| \mathbf{X}_1, \ldots, \mathbf{X}_n \right] \right]. \quad (7)$$

As for the conditional expectation in (7), notice that by independence

$$\mathbb{E} \left[ \left\| \sum_{i=1}^{n} r_i \mathbf{X}_i \right\|^p \middle| \mathbf{X}_1 = \mathbf{x}_1, \ldots, \mathbf{X}_n = \mathbf{x}_n \right] = \mathbb{E} \left[ \left\| \sum_{i=1}^{n} r_i \mathbf{x}_i \right\|^p \right] \quad (8)$$

According to Lemma 8, for $v_n$-almost every $\mathbf{x}_1, \ldots, \mathbf{x}_n \in \mathcal{X}^n$, where $v_n := \mathbb{P} \circ (\mathbf{X}_1, \ldots, \mathbf{X}_n)^{-1}$ denotes the distribution of $(\mathbf{X}_1, \ldots, \mathbf{X}_n)$, we have

$$\left[ \mathbb{E} \left\| \sum_{i=1}^{n} r_i \mathbf{x}_i \right\|^p \right] \leq C \cdot \left[ \sum_{i=1}^{n} \|\mathbf{x}_i\|^2 \right]^{\frac{p}{2}}, \quad (9)$$

where $C = 2^{\frac{p}{2}} \frac{\Gamma\left(\frac{p+1}{2}\right)}{\sqrt{\pi}}$ and $C$ is optimal. This means that for any constant $C'$ such that

$$\left[\mathbb{E} \left\|\sum_{i=1}^{n} r_i \mathbf{x}_i \right\|^p \right] \leq C' \cdot \left[\sum_{i=1}^{n} \|\mathbf{x}_i\|^2 \right]^{\frac{p}{2}}, \tag{10}$$

for all $n \in \mathbb{N}$ and for each collection of vectors $\mathbf{x}_1, \ldots, \mathbf{x}_n$, it follows that $C' \geq C$.

From (8) and (9), we can infer that

$$\mathbb{E} \left[\left\|\sum_{i=1}^{n} r_i \mathbf{X}_i \right\|^p \, \middle| \, \mathbf{X}_1 = \mathbf{x}_1, \ldots, \mathbf{X}_n = \mathbf{x}_n \right] \leq C \cdot \left[\sum_{i=1}^{n} \|\mathbf{X}_i\|^2 \right]^{\frac{p}{2}}.$$

Taking expectations in the above inequalities and (7) yield that

$$\mathbb{E} \left[\left\|\sum_{i=1}^{n} \mathbf{X}_i \right\|^p \right] \leq C \cdot \mathbb{E} \left[\sum_{i=1}^{n} \|\mathbf{X}_i\|^2 \right]^{\frac{p}{2}}. \tag{11}$$

To see optimality let the above statement hold for some constants $C'$ in place of $C$. Then if we choose $\mathbf{X}_i := \mathbf{x}_i r_i, 1 \leq i \leq n$ with arbitrary reals vectors $\mathbf{x}_1, \ldots, \mathbf{x}_n$, it follows that

$$\mathbb{E} \left[\left\|\sum_{i=1}^{n} r_i \mathbf{x}_i \right\|^p \right] \leq C' \cdot \mathbb{E} \left[\sum_{i=1}^{n} \|\mathbf{x}_i\|^2 \right]^{\frac{p}{2}},$$

whence we can conclude from (10) that $C' \geq C$. Thus we obtain that $C' = C$.

Notice that by Holder's inequality

$$\left[\sum_{i=1}^{n} \|\mathbf{X}_i\|^2 \right]^{\frac{p}{2}} \leq n^{p/2-1} \sum_{i=1}^{n} \|\mathbf{X}_i\|^p. \tag{12}$$

Plugging (12) into (11), we have

$$\mathbb{E} \left[\left\|\sum_{i=1}^{n} \mathbf{X}_i \right\|^p \right] \leq C \cdot 2^p n^{p/2-1} \cdot \mathbb{E} \left[\sum_{i=1}^{n} \|\mathbf{X}_i\|^p \right],$$

where $C = 2^{\frac{p}{2}} \frac{\Gamma\left(\frac{p+1}{2}\right)}{\sqrt{\pi}}$ is a constant.

Next, we use the following form of Stirling's formula for the Gamma-function, which follows from (6.1.5), (6.1.15) and (6.1.38) in Davis (1972) to bound the constant $C$. For every $x > 0$, there exists a $\mu(x) \in (0, 1/(12x))$ such that

$$\Gamma(x) = \sqrt{2\pi} x^{x-1/2} e^{-x} e^{\mu(x)}.$$

Thus

$$C = 2^{\frac{p}{2}} \frac{\Gamma\left(\frac{p+1}{2}\right)}{\sqrt{\pi}} = g(p) \sqrt{2} e^{-p/2} p^{p/2},$$

with $g(p) = \left(1 + \frac{1}{p}\right)^{p/2} e^{v(p)-1/2}$, where $0 < v(p) < 1/(6(p+1))$. By Taylor's formula we have that

$$\log(1 + x) = \sum_{m=1}^{\infty} \frac{1}{m} (-1)^{m-1} x^m, \quad \forall x \in (-1, 1],$$

and that for every $k \in \mathbb{N}_0$

$$\sum_{m=1}^{2k} \frac{1}{m} (-1)^{m-1} x^m \leq \log(1 + x) \leq \sum_{m=1}^{2k+1} \frac{1}{m} (-1)^{m-1} x^m, \forall x \geq 0.$$

Therefor we obtain with $k = 1$ that

$$\log g(p) = \frac{p}{2} \log(1 + \frac{1}{p}) + v(p) - \frac{1}{2} \leq -\frac{1}{4p} + \frac{1}{6p^2} + \frac{1}{6(p+1)} \leq -\frac{1}{18p},$$

where the last equality follows from elementary calculus. Similarly,

$$\log g(p) = \frac{p}{2} \log(1 + \frac{1}{p}) + v(p) - \frac{1}{2} \geq -\frac{1}{4p} + v(p) \geq -\frac{1}{4p},$$

Thus, we have

$$e^{-\frac{1}{4p}} \sqrt{2} e^{-p/2} p^{p/2} < C < e^{-\frac{1}{18p}} \sqrt{2} e^{-p/2} p^{p/2},$$

which implies that $C$ is strictly smaller than $\sqrt{2} e^{-p/2} p^{p/2}$ for all $p \geq 2$.

Since $C = \frac{1}{g(p)} \sqrt{2} e^{-p/2} p^{p/2}$ and $g(p) \geq e^{-\frac{1}{4p}}$ , we can obtain that the relative error between $C$ and $\sqrt{2} e^{-p/2} p^{p/2}$ is equal to

$$\frac{1}{g(p)} - 1 \leq e^{-\frac{1}{4p}} - 1 \leq \frac{1}{4p} e^{\frac{1}{4p}}$$

using Mean Value Theorem. This implies that the corresponding relative errors between $C$ and $\sqrt{2} e^{-p/2} p^{p/2}$ converge to zero as $p$ tends to infinity.

The proof is complete.

$\square$

Then we introduce the McDiarmid's inequality for vector-valued functions. We firstly consider real-valued functions, which follows from the standard tail-bound of McDiarmid's inequality and Proposition 2.5.2 in Vershynin (2018).

**Lemma 9** (McDiarmid's Inequality for real-valued functions). *Let $Z_i, \ldots, Z_n$ be independent random variables, and $f : \mathcal{Z}^n \mapsto \mathbb{R}$ such that the following inequality holds for any $z_i, \ldots, z_{i-1}, z_{i+1}, \ldots, z_n$*

$$\sup_{z_i, z_i'} |f(z_1, \ldots, z_{i-1}, z_i, z_{i+1}, \ldots, z_n) - f(z_1, \ldots, z_{i-1}, z_i', z_{i+1}, \ldots, z_n)| \leq \beta,$$

*Then for any $p > 1$ we have*

$$\|f(Z_1, \ldots, Z_n) - \mathbb{E}f(Z_1, \ldots, Z_n)\|_p \leq \sqrt{2pn} \beta.$$

To derive the McDiarmid's inequality for vector-valued functions, we need the expected distance between $\mathbf{f}(Z_1, \ldots, Z_n)$ and its expectation.

**Lemma 10** ((Rivasplata et al., 2018)). *Let $Z_i, \ldots, Z_n$ be independent random variables, and $\mathbf{f} : \mathcal{Z}^n \mapsto \mathcal{H}$ is a function into a Hilbert space $\mathcal{H}$ such that the following inequality holds for any $z_i, \ldots, z_{i-1}, z_{i+1}, \ldots, z_n$*

$$\sup_{z_i, z_i'} \|\mathbf{f}(z_1, \ldots, z_{i-1}, z_i, z_{i+1}, \ldots, z_n) - \mathbf{f}(z_1, \ldots, z_{i-1}, z_i', z_{i+1}, \ldots, z_n)\| \leq \beta,$$

*Then we have*

$$\mathbb{E}\left[\|\mathbf{f}(Z_1, \ldots, Z_n) - \mathbb{E}\mathbf{f}(Z_1, \ldots, Z_n)\|\right] \leq \sqrt{n} \beta.$$

Now, we can easily derive the $p$-norm McDiarmid's inequality for vector-valued functions which refines from Fan & Lei (2024) with better constants.

**Lemma 11** (McDiarmid's inequality for vector-valued functions). *Let $Z_i, \ldots, Z_n$ be independent random variables, and $\mathbf{f} : \mathcal{Z}^n \mapsto \mathcal{H}$ is a function into a Hilbert space $\mathcal{H}$ such that the following inequality holds for any $z_i, \ldots, z_{i-1}, z_{i+1}, \ldots, z_n$*

$$\sup_{z_i, z_i'} \|\mathbf{f}(z_1, \ldots, z_{i-1}, z_i, z_{i+1}, \ldots, z_n) - \mathbf{f}(z_1, \ldots, z_{i-1}, z_i', z_{i+1}, \ldots, z_n)\| \leq \beta, \tag{13}$$

*Then for any $p > 1$ we have*

$$\|\|\mathbf{f}(Z_1, \ldots, Z_n) - \mathbb{E}\mathbf{f}(Z_1, \ldots, Z_n)\|\|_p \leq (\sqrt{2p} + 1)\sqrt{n} \beta.$$

*Proof of Lemma 11.* Define a real-valued function $h : \mathcal{Z}^n \mapsto \mathbb{R}$ as

$$h(z_1, \ldots, z_n) = \|\mathbf{f}(z_1, \ldots, z_n) - \mathbb{E}[\mathbf{f}(Z_1, \ldots, Z_n)]\|.$$

We notice that this function satisfies the increment condition. For any $i$ and $z_1, \ldots, z_{i-1}, z_{i+1}, \ldots, z_n$, we have

$$\sup_{z_i, z_i'} |h(z_1, \ldots, z_{i-1}, z_i, z_{i+1}, \ldots, z_n) - h(z_1, \ldots, z_{i-1}, z_i', z_{i+1}, \ldots, z_n)|$$

$$= \sup_{z_i, z_i'} |\|\mathbf{f}(z_1, \ldots, z_n) - \mathbb{E}[\mathbf{f}(Z_1, \ldots, Z_n)]\| - \|\mathbf{f}(z_1, \ldots, z_{i-1}, z_i', z_{i+1}, \ldots, z_n) - \mathbb{E}[\mathbf{f}(Z_1, \ldots, Z_n)]\||$$

$$\leq \sup_{z_i, z_i'} |\|\mathbf{f}(z_1, \ldots, z_n) - \mathbf{f}(z_1, \ldots, z_{i-1}, z_i', z_{i+1}, \ldots, z_n)\| \leq \beta.$$

Therefore, we can apply Lemma 9 to the real-valued function $h$ and derive the following inequality

$$\|h(Z_1, \ldots, Z_n) - \mathbb{E}[h(Z_1, \ldots, Z_n)]\|_p \leq \sqrt{2pn}\beta.$$

According to Lemma 10, we know the following inequality $\mathbb{E}[h(Z_1, \ldots, Z_n)] \leq \sqrt{n}\beta$. Combing the above two inequalities together and we can derive the following inequality

$$\|\|\mathbf{f}(Z_1, \ldots, Z_n) - \mathbb{E}\mathbf{f}(Z_1, \ldots, Z_n)\|\|_p$$
$$\leq \|h(Z_1, \ldots, Z_n) - \mathbb{E}[h(Z_1, \ldots, Z_n)]\|_p + \|\mathbb{E}[h(Z_1, \ldots, Z_n)]\|_p$$
$$\leq (\sqrt{2p} + 1)\sqrt{n}\beta.$$

The proof is complete.

$\square$

*Proof of Theorem 1.* For $\mathbf{g}(Z_1, \ldots, Z_n)$ and $A \subset [n]$, we write $\|\|\mathbf{g}\|\|_p(Z_A) = (\mathbb{E}[\|f\|^p \, Z_A])^{\frac{1}{p}}$. Without loss of generality, we suppose that $n = 2^k$. Otherwise, we can add extra functions equal to zero, increasing the number of therms by at most two times.

Consider a sequence of partitions $\mathcal{P}_0, \ldots, \mathcal{P}_k$ with $\mathcal{P}_0 = \{\{i\} : i \in [n]\}, \mathcal{P}_k$ with $\mathcal{P}_n = [n]$, and to get $\mathcal{P}_l$ from $\mathcal{P}_{l+1}$ we split each subset in $\mathcal{P}_{l+1}$ into two equal parts. We have

$$\mathcal{P}_0 = \{\{1\}, \ldots, \{2^k\}\}, \quad \mathcal{P}_1 = \{\{1, 2\}, \{3, 4\}, \ldots, \{2^k - 1, 2^k\}\}, \quad \mathcal{P}_k = \{\{1, \ldots, 2^k\}\}.$$

We have $|\mathcal{P}_l| = 2^{k-l}$ and $|P| = 2^l$ for each $P \in \mathcal{P}_l$. For each $i \in [n]$ and $l = 0, \ldots, k$, denote by $P^l(i) \in \mathcal{P}_l$ the only set from $\mathcal{P}_l$ that contains $i$. In particular, $P^0(i) = \{i\}$ and $P^K(i) = [n]$.

For each $i \in [n]$ and every $l = 0, \ldots, k$ consider the random variables

$$\mathbf{g}_i^l = \mathbf{g}_i^l(Z_i, Z_{[n] \setminus P^l(i)}) = \mathbb{E}[\mathbf{g}_i | Z_i, Z_{[n] \setminus P^l(i)}],$$

i.e. conditioned on $z_i$ and all the variables that are not in the same set as $Z_i$ in the partition $\mathcal{P}_l$. In particular, $\mathbf{g}_i^0 = \mathbf{g}_i$ and $\mathbf{g}_i^k = \mathbb{E}[\mathbf{g}_i | Z_i]$. We can write a telescopic sum for each $i \in [n]$,

$$\mathbf{g}_i - \mathbb{E}[\mathbf{g}_i | Z_i] = \sum_{l=1}^{k-1} \mathbf{g}_i^l - \mathbf{g}_i^{l+1}.$$

Then, by the triangle inequality

$$\left\|\left\|\sum_{i=1}^n \mathbf{g}_i\right\|\right\|_p \leq \left\|\left\|\sum_{i=1}^n \mathbb{E}[\mathbf{g}_i | Z_i]\right\|\right\|_p + \sum_{l=0}^{k-1} \left\|\left\|\sum_{i=1}^n \mathbf{g}_i^l - \mathbf{g}_i^{l+1}\right\|\right\|_p. \tag{14}$$

To bound the first term, since $\|\mathbb{E}[\mathbf{g}_i | Z_i]\| \leq G$, we can check that the vector-valued function $\mathbf{f}(Z_1, \ldots, Z_n) = \sum_{i=1}^n \mathbb{E}[\mathbf{g}_i | Z_i]$ satisfies (13) with $\beta = 2G$, and $\mathbb{E}[\mathbb{E}[\mathbf{g}_i | Z_i]] = 0$, applying Lemma 11 with $\beta = 2G$, we have

$$\left\|\left\|\sum_{i=1}^n \mathbb{E}[\mathbf{g}_i | Z_i]\right\|\right\|_p \leq 2(\sqrt{2p} + 1)\sqrt{n}G. \tag{15}$$

Then we start to bound the second term of the right hand side of (14). Observe that

$$\mathbf{g}_i^{l+1}(Z_i, Z_{[n]\setminus P^{l+1}(i)}) = \mathbb{E}\left[\mathbf{g}_i^l(Z_i, Z_{[n]\setminus P^l(i)}) \big| Z_i, Z_{[n]\setminus P^{l+1}(i)}\right],$$

where the expectation is taken with respect to the variables $Z_j, j \in P^{l+1}(i)\setminus P^l(i)$. Changing any $Z_j$ would change $\mathbf{g}_i^l$ by $\beta$. Therefore, we apply Lemma 11 with $\mathbf{f} = \mathbf{g}_i^l$ where there are $2^l$ random variables and obtain a uniform bound

$$\left\|\left\|\mathbf{g}_i^l - \mathbf{g}_i^{l+1}\right\|\right\|_p (Z_i, Z_{[n]\setminus P^{l+1}(i)}) \le (\sqrt{2p} + 1)\sqrt{2^l}\beta, \quad \forall p \ge 2,$$

Taking integration over $(Z_i, Z_{[n]\setminus P^{l+1}(i)})$, we have $\left\|\left\|\mathbf{g}_i^l - \mathbf{g}_i^{l+1}\right\|\right\|_p \le (\sqrt{2p} + 1)\sqrt{2^l}\beta$ as well.

Next, we turn to the sum $\sum_{i \in P^l} \mathbf{g}_i^l - \mathbf{g}_i^{l+1}$ for any $P^l \in \mathcal{P}_l$. Since $\mathbf{g}_i^l - \mathbf{g}_i^{l+1}$ for $i \in P^l$ depends only on $Z_i, Z_{[n]\setminus P^l}$, the terms are independent and zero mean conditioned on $Z_{[n]\setminus P^l}$. Applying Lemma 7, we have for any $p \ge 2$,

$$\left\|\left\|\sum_{i \in P^l} \mathbf{g}_i^l - \mathbf{g}_i^{l+1}\right\|\right\|_p^p (Z_{[n]\setminus P^l}) \le \left(2 \cdot 2^{\frac{1}{2p}} \sqrt{\frac{2^l p}{e}}\right)^p \frac{1}{2^l} \sum_{i \in P^l} \left\|\left\|\mathbf{g}_i^l - \mathbf{g}_i^{l+1}\right\|\right\|_p^p (Z_{[n]\setminus P^l}).$$

Integrating with respect to $(Z_{[n]\setminus P^l})$ and using $\left\|\left\|\mathbf{g}_i^l - \mathbf{g}_i^{l+1}\right\|\right\|_p \le (\sqrt{2p} + 1)\sqrt{2^l}\beta$, we have

$$\left\|\left\|\sum_{i \in P^l} \mathbf{g}_i^l - \mathbf{g}_i^{l+1}\right\|\right\|_p \le \left(2 \cdot 2^{\frac{1}{2p}} \sqrt{\frac{2^l p}{e}}\right) \frac{1}{2^l} \times 2^l(\sqrt{2p} + 1)\sqrt{2^l}\beta$$

$$= 2^{1+\frac{1}{2p}} \left(\sqrt{\frac{p}{e}}\right)(\sqrt{2p} + 1)2^l\beta.$$

Then using triangle inequality over all sets $P^l \in \mathcal{P}_l$, we have

$$\left\|\left\|\sum_{i \in [n]} \mathbf{g}_i^l - \mathbf{g}_i^{l+1}\right\|\right\|_p \le \sum_{P^l \in \mathcal{P}_l} \left\|\left\|\sum_{i \in P^l} \mathbf{g}_i^l - \mathbf{g}_i^{l+1}\right\|\right\|_p$$

$$\le 2^{k-l} \times 2^{1+\frac{1}{2p}} \left(\sqrt{\frac{p}{e}}\right)(\sqrt{2p} + 1)2^l\beta$$

$$\le 2^{1+\frac{1}{2p}} \left(\sqrt{\frac{p}{e}}\right)(\sqrt{2p} + 1)2^k\beta.$$

Recall that $2^k \le n$ due to the possible extension of the sample. Then we have

$$\sum_{l=0}^{k-1} \left\|\left\|\sum_{i=1}^n \mathbf{g}_i^l - \mathbf{g}_i^{i+1}\right\|\right\|_p \le 2^{2+\frac{1}{2p}} \left(\sqrt{\frac{p}{e}}\right)(\sqrt{2p} + 1)n\beta \lceil \log_2 n \rceil.$$

We can plug the above bound together with (15) into (14), to derive the following inequality

$$\left\|\left\|\sum_{i=1}^n \mathbf{g}_i\right\|\right\|_p \le 2(\sqrt{2p} + 1)\sqrt{n}G + 2^{2+\frac{1}{2p}} \left(\sqrt{\frac{p}{e}}\right)(\sqrt{2p} + 1)n\beta \lceil \log_2 n \rceil.$$

The proof is completed.

$\square$

## C.2 PROOFS OF SUBSECTION 3.2

*Proof of Theorem 2.* Let $S = \{z_1, \ldots, z_n\}$ be a set of independent random variables each taking values in $\mathcal{Z}$ and $S' = \{z_1', \ldots, z_n'\}$ be its independent copy. For any $i \in [n]$, define $S^{(i)} =$

$\{z_i, \ldots, z_{i-1}, z_i', z_{i+1}, \ldots, z_n\}$ be a dataset replacing the $i$-th sample in $S$ with another i.i.d. sample $z_i'$. Then we can firstly write the following decomposition

$$n\nabla F(A(S)) - n\nabla F_S(A(S))$$

$$= \sum_{i=1}^{n} \mathbb{E}_Z \left[ \nabla f(A(S); Z)] - \mathbb{E}_{z_i'} \left[ \nabla f(A(S^{(i)}), Z) \right] \right]$$

$$+ \sum_{i=1}^{n} \mathbb{E}_{z_i'} \left[ \mathbb{E}_Z \left[ \nabla f(A(S^{(i)}), Z) \right] - \nabla f(A(S^{(i)}), z_i) \right]$$

$$+ \sum_{i=1}^{n} \mathbb{E}_{z_i'} \left[ \nabla f(A(S^{(i)}), z_i) \right] - \sum_{i=1}^{n} \nabla f(A(S), z_i).$$

We denote that $\mathbf{g}_i(S) = \mathbb{E}_{z_i'} \left[ \mathbb{E}_Z \left[ \nabla f(A(S^{(i)}), Z) \right] - \nabla f(A(S^{(i)}), z_i) \right]$, thus we have

$$\|n\nabla F(A(S)) - n\nabla F_S(A(S))\|_2$$

$$= \left\| \sum_{i=1}^{n} \mathbb{E}_Z \left[ \nabla f(A(S); Z)] - \mathbb{E}_{z_i'} \left[ \nabla f(A(S^{(i)}), Z) \right] \right] \right.$$

$$+ \sum_{i=1}^{n} \mathbb{E}_{z_i'} \left[ \mathbb{E}_Z \left[ \nabla f(A(S^{(i)}), Z) \right] - \nabla f(A(S^{(i)}), z_i) \right] \quad (16)$$

$$\left. + \sum_{i=1}^{n} \mathbb{E}_{z_i'} \left[ \nabla f(A(S^{(i)}), z_i) \right] - \sum_{i=1}^{n} \nabla f(A(S), z_i) \right\|_2$$

$$\leq 2n\beta + \left\| \sum_{i=1}^{n} \mathbf{g}_i(S) \right\|_2,$$

where the inequality holds from the definition of uniform stability in gradients.

According to our assumptions, we get $\|\mathbf{g}_i(S)\|_2 \leq 2M$ and

$$\mathbb{E}_{z_i}[\mathbf{g}_i(S)] = \mathbb{E}_{z_i}\mathbb{E}_{z_i'} \left[ \mathbb{E}_Z \left[ \nabla f(A(S^{(i)}); Z) \right] - \nabla f(A(S^{(i)}); z_i) \right]$$

$$= \mathbb{E}_{z_i'} \left[ \mathbb{E}_Z \left[ \nabla f(A(S^{(i)}); Z) \right] - \mathbb{E}_{z_i} \left[ \nabla f(A(S^{(i)}); z_i) \right] \right] = 0,$$

where this equality holds from the fact that $z_i$ and $Z$ follow from the same distribution. For any $i \in [n]$, any $j \neq i$ and any $z_j''$, we have

$$\left\| \mathbf{g}_i(z_1, \ldots, z_{j-1}, z_j, z_{j+1}, \ldots, z_n) - \mathbf{g}_i(z_1, \ldots, z_{j-1}, z_j'', z_{j+1}, \ldots, z_n) \right\|_2$$

$$\leq \left\| \mathbb{E}_{z_i'} \left[ \mathbb{E}_Z \left[ \nabla f(A(S^{(i)}); Z) - \nabla f(A(S^{(i)}); z_i) \right] - \mathbb{E}_{z_i'} \left[ \mathbb{E}_Z \left[ \nabla f(A(S_j^{(i)}); Z) - \nabla f(A(S_j^{(i)}); z_i) \right] \right] \right\|_2$$

$$\leq \left\| \mathbb{E}_{z_i'} \left[ \mathbb{E}_Z \left[ \nabla f(A(S^{(i)}); Z) - \nabla f(A(S_j^{(i)}); Z) \right] \right] \right\|_2 + \left\| \mathbb{E}_{z_i'} \left[ \mathbb{E}_Z \left[ \nabla f(A(S^{(i)}); Z) \right] - \nabla f(A(S_j^{(i)}); z_i) \right] \right\|_2$$

$$\leq 2\beta,$$

where $S^{(i)} = \{z_i, \ldots, z_{i-1}, z_i', z_{i+1}, \ldots, z_n\}$. Thus, we have verified that three conditions in Theorem 1 are satisfied for $\mathbf{g}_i(S)$. We have the following result for any $p > 2$

$$\left\| \left\| \sum_{i=1}^{n} \mathbf{g}_i(S) \right\| \right\|_p \leq 4(\sqrt{2p} + 1)\sqrt{n}M + 8 \times 2^{\frac{1}{4}} \left( \sqrt{\frac{p}{e}} \right) (\sqrt{2p} + 1)n\beta \lceil \log_2 n \rceil.$$

We can combine the above inequality and (16) to derive the following inequality

$$n \|\|\nabla F(A(S)) - n\nabla F_S(A(S))\|\|_p$$

$$\leq 2n\beta + 4(\sqrt{2p} + 1)\sqrt{n}M + 8 \times 2^{\frac{1}{4}} \left( \sqrt{\frac{p}{e}} \right) (\sqrt{2p} + 1)n\beta \lceil \log_2 n \rceil.$$

According to Lemma 4 for any $\delta \in (0, 1)$, with probability at least $1 - \delta$, we have

$$n\|\nabla F(A(S)) - \nabla F_S(A(S))\|_2$$

$$\leq 2n\beta + 4\sqrt{n}M + 8 \times 2^{\frac{3}{4}}\sqrt{e}n\beta \lceil \log_2 n \rceil \log(e/\delta) + (4e\sqrt{2n}M + 8 \times 2^{\frac{1}{4}}\sqrt{e}n\beta \lceil \log_2 n \rceil)\sqrt{\log e/\delta}.$$

This implies that

$$\|\nabla F(A(S)) - \nabla F_S(A(S))\|_2$$

$$\leq 2\beta + \frac{4M\left(1 + e\sqrt{2\log(e/\delta)}\right)}{\sqrt{n}} + 8 \times 2^{\frac{1}{4}}(\sqrt{2} + 1)\sqrt{e}\beta \lceil \log_2 n \rceil \log(e/\delta).$$

The proof is completed. $\qquad\square$

*Proof of Theorem 3.* We can firstly write the following decomposition

$$n\nabla F(A(S)) - n\nabla F_S(A(S))$$

$$= \sum_{i=1}^{n} \mathbb{E}_Z\left[\nabla f(A(S); Z)] - \mathbb{E}_{z_i'}\left[\nabla f(A(S^{(i)}), Z)\right]\right]$$

$$+ \sum_{i=1}^{n} \mathbb{E}_{z_i'}\left[\mathbb{E}_Z\left[\nabla f(A(S^{(i)}), Z)\right] - \nabla f(A(S^{(i)}), z_i)\right]$$

$$+ \sum_{i=1}^{n} \mathbb{E}_{z_i'}\left[\nabla f(A(S^{(i)}), z_i)\right] - \sum_{i=1}^{n} \nabla f(A(S), z_i).$$

We denote that $\mathbf{h}_i(S) = \mathbb{E}_{z_i'}\left[\mathbb{E}_Z\left[\nabla f(A(S^{(i)}), Z)\right] - \nabla f(A(S^{(i)}), z_i)\right]$, we have

$$n\nabla F(A(S)) - n\nabla F_S(A(S)) - \sum_{i=1}^{n} \mathbf{h}_i(S)$$

$$= \sum_{i=1}^{n} \mathbb{E}_Z\left[\nabla f(A(S); Z)] - \mathbb{E}_{z_i'}\left[\nabla f(A(S^{(i)}), Z)\right]\right]$$

$$+ \sum_{i=1}^{n} \mathbb{E}_{z_i'}\left[\nabla f(A(S^{(i)}), z_i)\right] - \sum_{i=1}^{n} \nabla f(A(S), z_i),$$

which implies that

$$\left\|n\nabla F(A(S)) - n\nabla F_S(A(S)) - \sum_{i=1}^{n} \mathbf{h}_i(S)\right\|_2$$

$$= \left\|\sum_{i=1}^{n} \mathbb{E}_Z\left[\nabla f(A(S); Z)] - \mathbb{E}_{z_i'}\left[\nabla f(A(S^{(i)}), Z)\right]\right]\right.$$

$$\left.+ \sum_{i=1}^{n} \mathbb{E}_{z_i'}\left[\nabla f(A(S^{(i)}), z_i)\right] - \sum_{i=1}^{n} \nabla f(A(S), z_i)\right\|_2$$

$$\leq 2n\beta,$$

$$(17)$$

where the inequality holds from the definition of uniform stability in gradients.

Then, for any $i = 1, \ldots, n$, we define $\mathbf{q}_i(S) = \mathbf{h}_i(S) - \mathbb{E}_{S\{z_i\}}[\mathbf{h}_i(S)]$. It is easy to verify that $\mathbb{E}_{S\backslash\{z_i\}}[\mathbf{q}_i(S)] = \mathbf{0}$ and $\mathbb{E}_{z_i}[\mathbf{h}_i(S)] = \mathbb{E}_{z_i}[\mathbf{q}_i(S)] - \mathbb{E}_{z_i}\mathbb{E}_{S\backslash\{z_i\}}[\mathbf{q}_i(S)] = \mathbf{0} - \mathbf{0} = \mathbf{0}$. Also, for any $j \in [n]$ with $j \neq i$ and $z_j'' \in \mathcal{Z}$, we have the following inequality

$$\|\mathbf{q}_i(S) - \mathbf{q}_i(z_1, \ldots, z_{j-1}, z_j'', z_{j+1}, \ldots, z_n)\|_2$$

$$\leq \|\mathbf{h}_i(S) - \mathbf{h}_i(z_1, \ldots, z_{j-1}, z_j'', z_{j+1}, \ldots, z_n)\|_2$$

$$+ \|\mathbb{E}_{S\backslash\{z_i\}}[\mathbf{h}_i(S)] - \mathbb{E}_{S\backslash\{z_i\}}[\mathbf{h}_i(1, \ldots, z_{j-1}, z_j'', z_{j+1}, \ldots, z_n)]\|_2.$$

For the first term $\|\mathbf{h}_i(S) - \mathbf{h}_i(z_1, \ldots, z_{j-1}, z_j'', z_{j+1}, \ldots, z_n)\|_2$, it can be bounded by $2\beta$ according to the definition of uniform stability. Similar result holds for the second term $\|\mathbb{E}_{S \setminus \{z_i\}}[\mathbf{h}_i(S)] - \mathbb{E}_{S \setminus \{z_i\}}[\mathbf{h}_i(z_1, \ldots, z_{j-1}, z_j'', z_{j+1}, \ldots, z_n)]\|_2$ according to the uniform stability. By a combination of the above analysis, we get $\|\mathbf{q}_i(S) - \mathbf{q}_i(z_1, \ldots, z_{j-1}, z_j'', z_{j+1}, \ldots, z_n)\|_2 \le \|\mathbf{h}_i(S) - \mathbf{h}_i(z_1, \ldots, z_{j-1}, z_j'', z_{j+1}, \ldots, z_n)\|_2 \le 4\beta$.

Thus, we have verified that three conditions in Theorem 1 are satisfied for $\mathbf{q}_i(S)$. We have the following result for any $p \ge 2$

$$\left\|\left\|\sum_{i=1}^n \mathbf{q}_i(S)\right\|\right\|_p \le 2^{4+\frac{1}{4}} \left(\sqrt{\frac{p}{e}}\right)(\sqrt{2p}+1)n\beta \lceil \log_2 n \rceil. \tag{18}$$

Furthermore, we can derive that

$$n\nabla F(A(S)) - n\nabla F_S(A(S)) - \sum_{i=1}^n \mathbf{h}_i(S) + \sum_{i=1}^n \mathbf{q}_i(S)$$

$$= n\nabla F(A(S)) - n\nabla F_S(A(S)) - \sum_{i=1}^n \mathbb{E}_{S \setminus \{z_i\}}[\mathbf{h}_i(S)]$$

$$= n\nabla F(A(S)) - n\nabla F_S(A(S)) - n\mathbb{E}_{S'}[\nabla F(A(S'))] + n\mathbb{E}_S[\nabla F(A(S))].$$

Due to the i.i.d. property between $S$ and $S'$, we know that $\mathbb{E}_{S'}[\nabla F(A(S'))] = \mathbb{E}_S[\nabla F(A(S))]$. Thus, combined above equality, (17) and (18), we have

$$\left\|\left\|n\nabla F(A(S)) - n\nabla F_S(A(S)) - n\mathbb{E}_S[\nabla F(A(S))] + n\mathbb{E}_{S'}[\nabla F_S(A(S'))]\right\|\right\|_p$$

$$\le \left\|\left\|n\nabla F(A(S)) - n\nabla F_S(A(S)) - \sum_{i=1}^n \mathbf{h}_i(S)\right\|\right\|_p$$

$$+ \left\|\left\|\sum_{i=1}^n \mathbf{h}_i(S) - n\mathbb{E}_S[\nabla F(A(S))] + n\mathbb{E}_{S'}F_S[A(S')]\right\|\right\|_p$$

$$= \left\|\left\|n\nabla F(A(S)) - n\nabla F_S(A(S)) - \sum_{i=1}^n \mathbf{h}_i(S)\right\|\right\|_p + \left\|\left\|\sum_{i=1}^n \mathbf{q}_i(S)\right\|\right\|_p$$

$$\le 2n\beta + 2^{4+\frac{1}{4}} \left(\sqrt{\frac{p}{e}}\right)(\sqrt{2p}+1)n\beta \lceil \log_2 n \rceil$$

$$\le 16 \times 2^{\frac{3}{4}} \left(\sqrt{\frac{1}{e}}\right) pn\beta \lceil \log_2 n \rceil + 32 \left(\sqrt{\frac{1}{e}}\right)\sqrt{p}n\beta \lceil \log_2 n \rceil.$$

According to Lemma 4 for any $\delta \in (0,1)$, with probability at least $1 - \delta/3$, we have

$$\|\nabla F(A(S)) - \nabla F_S(A(S))\|_2$$

$$\le \|\mathbb{E}_{S'}[\nabla F_S(A(S'))] - \mathbb{E}_S[\nabla F(A(S))]\|_2 \tag{19}$$

$$+ 16 \times 2^{\frac{3}{4}}\sqrt{e}\beta \lceil \log_2 n \rceil \log(3e/\delta) + 32\sqrt{e}\beta \lceil \log_2 n \rceil \sqrt{\log 3e/\delta}.$$

Next, we need to bound the term $\|\mathbb{E}_{S'}[\nabla F_S(A(S'))] - \mathbb{E}_S[\nabla F(A(S))]\|_2$. There holds that $\|\mathbb{E}_S\mathbb{E}_{S'}[\nabla F_S(A(S'))]\|_2 = \|\mathbb{E}_S[\nabla F(A(S))]\|_2$. Then, by the Bernstein inequality in Lemma 5, we obtain the following inequality with probability at least $1 - \delta/3$,

$$\|\mathbb{E}_{S'}[\nabla F_S(A(S'))] - \mathbb{E}_S[\nabla F(A(S))]\|_2 \le \sqrt{\frac{2\mathbb{E}_{z_i}[\|\mathbb{E}_{S'}\nabla f(A(S');z_i)\|_2^2]\log\frac{6}{\delta}}{n}} + \frac{M\log\frac{6}{\delta}}{n}. \tag{20}$$

Then using Jensen's inequality, we have

$$\mathbb{E}_{z_i}[\|\mathbb{E}_{S'}\nabla f(A(S');z_i)\|_2^2] \le \mathbb{E}_{z_i}\mathbb{E}_{S'}\|\nabla f(A(S');z_i)\|_2^2$$

$$= \mathbb{E}_Z\mathbb{E}_{S'}\|\nabla f(A(S');Z)\|_2^2 = \mathbb{E}_Z\mathbb{E}_S\|\nabla f(A(S);Z)\|_2^2. \tag{21}$$

Combing (19), (20) with (21), we finally obtain that with probability at least $1 - 2\delta/3$,

$$\|\nabla F(A(S)) - \nabla F_S(A(S))\|_2$$

$$\leq \sqrt{\frac{2\mathbb{E}_Z \mathbb{E}_S \|\nabla f(A(S); Z)\|_2^2 \log \frac{6}{\delta}}{n}} + \frac{M \log \frac{6}{\delta}}{n} \tag{22}$$

$$+ 16 \times 2^{\frac{3}{4}} \sqrt{e} \beta \lceil \log_2 n \rceil \log (3e/\delta) + 32\sqrt{e}\beta \lceil \log_2 n \rceil \sqrt{\log 3e/\delta}.$$

Next, since $S = \{z_i, \ldots, z_n\}$, we define $p = p(z_1, \ldots, z_n) = \mathbb{E}_Z[\|\nabla f(A(S); Z)\|_2^2]$ and $p_i = p_i(z_1, \ldots, z_n) = \sup_{z_i \in \mathcal{Z}} p(z_i, \ldots, z_n)$. So there holds $p_i \geq p$ for any $i = 1, \ldots, n$ and any $\{z_1, \ldots, z_n\} \in \mathcal{Z}^n$. Also, there holds that

$$\sum_{i=1}^{n} (p_i - p)^2$$

$$= \sum_{i=1}^{n} \left( \sup_{z_i \in \mathcal{Z}} \mathbb{E}_Z[\|\nabla f(A(S'); Z)\|_2^2] - \mathbb{E}_Z[\|\nabla f(A(S); Z)\|_2^2] \right)^2$$

$$\leq \sum_{i=1}^{n} \left( \mathbb{E}_Z \left[ \sup_{z_i \in \mathcal{Z}} \|\nabla f(A(S'); Z)\|_2^2 - \|\nabla f(A(S); Z)\|_2^2 \right] \right)^2$$

$$= \sum_{i=1}^{n} \left( \mathbb{E}_Z \left[ \left( \sup_{z_i \in \mathcal{Z}} \|\nabla f(A(S'); Z)\|_2 - \|\nabla f(A(S); Z)\|_2 \right) \left( \sup_{z_i \in \mathcal{Z}} \|\nabla f(A(S'); Z)\|_2 + \|\nabla f(A(S); Z)\|_2 \right) \right] \right)^2$$

$$\leq \sum_{i=1}^{n} \beta^2 \left( \mathbb{E}_Z \left[ \|\nabla f(A(S); Z)\|_2 + \sup_{z_i \in \mathcal{Z}} \|\nabla f(A(S); Z)\|_2 \right] \right)^2$$

$$\leq n\beta^2 \left( 2\mathbb{E}_Z[\|\nabla f(A(S); Z)\|_2 + \beta] \right)^2$$

$$\leq 8n\beta^2 p + 2n\beta^4, \tag{23}$$

where the first inequality follows from the Jensen's inequality. The second and third inequalities follow from the definition of uniform stability in gradients. The last inequality holds from that $(a + b)^2 \leq 2a^2 + 2b^2$.

From (23), we know that $p$ is $(8n\beta^2, 2n\beta^4)$ weakly self-bounded. Thus, by Lemma 6, we obtain that with probability at least $1 - \delta/3$,

$$\mathbb{E}_Z \mathbb{E}_S[\|\nabla f(A(S); Z)\|_2^2] - \mathbb{E}_Z[\|\nabla f(A(S); Z)\|_2^2]$$

$$\leq \sqrt{(16n\beta^2 \mathbb{E}_S \mathbb{E}_Z[\|\nabla f(A(S); Z)\|_2^2] + 4n\beta^4) \log(3/\delta)}$$

$$= \sqrt{(\mathbb{E}_S \mathbb{E}_Z[\|\nabla f(A(S); Z)\|_2^2] + \frac{1}{4}\beta^2) 16n\beta^2 \log(3/\delta)}$$

$$\leq \frac{1}{2} (\mathbb{E}_S \mathbb{E}_Z[\|\nabla f(A(S); Z)\|_2^2] + \frac{1}{4}\beta^2) + 8n\beta^2 \log(3/\delta),$$

where the last inequality follows from that $\sqrt{ab} \leq \frac{a+b}{2}$ for all $a, b > 0$. Thus, we have

$$\mathbb{E}_Z \mathbb{E}_S[\|\nabla f(A(S); Z)\|_2^2] \leq 2\mathbb{E}_Z[\|\nabla f(A(S); Z)\|_2^2] + \frac{1}{4}\beta^2 + 16n\beta^2 \log(3/\delta). \tag{24}$$

Substituting (24) into (22), we finally obtain that with probability at least $1 - \delta$

$$\|\nabla F(A(S)) - \nabla F_S(A(S))\|_2$$

$$\leq \sqrt{\frac{2 \left( 2\mathbb{E}_Z[\|\nabla f(A(S); Z)\|_2^2] + \frac{1}{4}\beta^2 + 16n\beta^2 \log(3/\delta) \right) \log \frac{6}{\delta}}{n}} + \frac{M \log \frac{6}{\delta}}{n} \tag{25}$$

$$+ 16 \times 2^{\frac{3}{4}} \sqrt{e} \beta \lceil \log_2 n \rceil \log (3e/\delta) + 32\sqrt{e}\beta \lceil \log_2 n \rceil \sqrt{\log 3e/\delta}.$$

According to inequality $\sqrt{a+b} = \sqrt{a} + \sqrt{b}$ for any $a, b > 0$, with probability at least $1 - \delta$, we have

$$\|\nabla F(A(S)) - \nabla F_S(A(S))\|_2$$

$$\leq \sqrt{\frac{4\mathbb{E}_Z[\|\nabla f(A(S); Z)\|_2^2] \log \frac{6}{\delta}}{n}} + \sqrt{\frac{\left(\frac{1}{2}\beta^2 + 32n\beta^2 \log(3/\delta)\right) \log \frac{6}{\delta}}{n}} + \frac{M \log \frac{6}{\delta}}{n}$$

$$+ 16 \times 2^{\frac{3}{4}} \sqrt{e}\beta \lceil \log_2 n \rceil \log(3e/\delta) + 32\sqrt{e}\beta \lceil \log_2 n \rceil \sqrt{\log 3e/\delta}.$$

The proof is complete.

$\square$

*Proof of Proposition 1.* According to the proof in Theorem 3, we have the following inequality with probability at least $1 - \delta$

$$\|\nabla F(A(S)) - \nabla F_S(A(S))\|_2$$

$$\leq \sqrt{\frac{2\left(2\mathbb{E}_Z[\|\nabla f(A(S); Z)\|_2^2] + \frac{1}{4}\beta^2 + 16n\beta^2 \log(3/\delta)\right) \log \frac{6}{\delta}}{n}} \tag{26}$$

$$+ \frac{M \log \frac{6}{\delta}}{n} + 16 \times 2^{\frac{3}{4}} \sqrt{e}\beta \lceil \log_2 n \rceil \log(3e/\delta) + 32\sqrt{e}\beta \lceil \log_2 n \rceil \sqrt{\log 3e/\delta}.$$

Since SGC implies that $\mathbb{E}_Z[\|\nabla f(\mathbf{w}; Z)\|_2^2] \leq \rho\|\nabla F(\mathbf{w})\|_2^2$, according to inequalities $\sqrt{ab} \leq \eta a + \frac{1}{\eta}b$ and $\sqrt{a+b} \leq \sqrt{a} + \sqrt{b}$ for any $a, b, \eta > 0$, we have the following inequality with probability at least $1 - \delta$

$$\|\nabla F(A(S)) - \nabla F_S(A(S))\|_2$$

$$\leq \sqrt{\frac{2\left(2\rho\|\nabla F(A(S))\|_2^2 + \frac{1}{4}\beta^2 + 16n\beta^2 \log(3/\delta)\right) \log \frac{6}{\delta}}{n}}$$

$$+ \frac{M \log \frac{6}{\delta}}{n} + 16 \times 2^{\frac{3}{4}} \sqrt{e}\beta \lceil \log_2 n \rceil \log(3e/\delta) + 32\sqrt{e}\beta \lceil \log_2 n \rceil \sqrt{\log 3e/\delta}$$

$$\leq \sqrt{\frac{\left(\frac{1}{2}\beta^2 + 32n\beta^2 \log(3/\delta)\right) \log \frac{6}{\delta}}{n}} + \frac{\eta}{1+\eta}\|\nabla F(A(S))\| + \frac{1+\eta}{\eta}\frac{4\rho M \log \frac{6}{\delta}}{n}$$

$$+ \frac{M \log \frac{6}{\delta}}{n} + 16 \times 2^{\frac{3}{4}} \sqrt{e}\beta \lceil \log_2 n \rceil \log(3e/\delta) + 32\sqrt{e}\beta \lceil \log_2 n \rceil \sqrt{\log 3e/\delta}.$$

which implies that

$$\|\nabla F(A(S))\|_2 \leq (1+\eta)\|\nabla F_S(A(S))\|_2 + C\frac{1+\eta}{\eta}\left(\frac{M}{n} \log \frac{6}{\delta} + \beta \log n \log \frac{1}{\delta}\right).$$

The proof is complete. $\square$

*Proof of Remark 7.* According to the proof in Theorem 3, we have the following inequality that with probability at least $1 - \delta$

$$\|\nabla F(A(S)) - \nabla F_S(A(S))\|_2$$

$$\leq \sqrt{\frac{4\mathbb{E}_Z[\|\nabla f(A(S); Z)\|_2^2] \log \frac{6}{\delta}}{n}} + \sqrt{\frac{\left(\frac{1}{2}\beta^2 + 32n\beta^2 \log(3/\delta)\right) \log \frac{6}{\delta}}{n}} + \frac{M \log \frac{6}{\delta}}{n} \tag{27}$$

$$+ 16 \times 2^{\frac{3}{4}} \sqrt{e}\beta \lceil \log_2 n \rceil \log(3e/\delta) + 32\sqrt{e}\beta \lceil \log_2 n \rceil \sqrt{\log 3e/\delta}.$$

Since $f(\mathbf{w})$ is $\gamma$-smooth, we have

$$\mathbb{E}_Z[\|\nabla f(A(S); Z)\|_2^2]$$

$$\leq \mathbb{E}_Z[\|\nabla f(A(S); Z) - \nabla f(\mathbf{w}^*; Z)\|_2^2 + \|\nabla f(\mathbf{w}^*; Z)\|_2^2] \tag{28}$$

$$\leq \gamma^2\|A(S) - \mathbf{w}^*\|_2^2 + \mathbb{E}_Z[\|\nabla f(\mathbf{w}^*; Z)\|_2^2]$$

Plugging (28) into (27), we have

$$\|\nabla F(A(S)) - \nabla F_S(A(S))\|_2$$

$$\leq \sqrt{\frac{4(\gamma^2 \|A(S) - \mathbf{w}^*\|_2^2 + \mathbb{E}_Z[\|\nabla f(\mathbf{w}^*; Z)\|_2^2]) \log \frac{6}{\delta}}{n}} + \sqrt{\frac{\left(\frac{1}{2}\beta^2 + 32n\beta^2 \log(3/\delta)\right) \log \frac{6}{\delta}}{n}}$$

$$+ \frac{M \log \frac{6}{\delta}}{n} + 16 \times 2^{\frac{3}{4}} \sqrt{e}\beta \lceil \log_2 n \rceil \log(3e/\delta) + 32\sqrt{e}\beta \lceil \log_2 n \rceil \sqrt{\log 3e/\delta}$$

$$\leq 2\gamma \|A(S) - \mathbf{w}^*\|_2 \sqrt{\frac{\log \frac{6}{\delta}}{n}} + \sqrt{\frac{4\mathbb{E}_Z[\|\nabla f(\mathbf{w}^*; Z)\|_2^2] \log \frac{6}{\delta}}{n}}$$

$$+ \sqrt{\frac{\left(\frac{1}{2}\beta^2 + 32n\beta^2 \log(3/\delta)\right) \log \frac{6}{\delta}}{n}} + \frac{M \log \frac{6}{\delta}}{n}$$

$$+ 16 \times 2^{\frac{3}{4}} \sqrt{e}\beta \lceil \log_2 n \rceil \log(3e/\delta) + 32\sqrt{e}\beta \lceil \log_2 n \rceil \sqrt{\log 3e/\delta},$$

(29)

where the second inequality holds because $\sqrt{a+b} \leq \sqrt{a} + \sqrt{b}$ for any $a, b > 0$, which means that

$$\|\nabla F(A(S)) - \nabla F_S(A(S))\|_2$$

$$\lesssim \beta \log n \log(1/\delta) + \frac{\log(1/\delta)}{n} + \sqrt{\frac{\mathbb{E}_Z\left[\nabla \|f(\mathbf{w}^*; Z)\|_2^2\right] \log(1/\delta)}{n}} + \|A(S) - \mathbf{w}^*\| \sqrt{\frac{\log(1/\delta)}{n}}.$$

The proof is complete.

$\square$

### C.3 PROOFS OF SUBSECTION 3.2

*Proof of Theorem 4.* Inequality (29) implies that

$$\|\nabla F(A(S))\|_2 - \|\nabla F_S(A(S))\|_2$$

$$\leq \sqrt{\frac{4(\gamma^2 \|A(S) - \mathbf{w}^*\|_2^2 + \mathbb{E}_Z[\|\nabla f(\mathbf{w}^*; Z)\|_2^2]) \log \frac{6}{\delta}}{n}} + \sqrt{\frac{\left(\frac{1}{2}\beta^2 + 32n\beta^2 \log(3/\delta)\right) \log \frac{6}{\delta}}{n}}$$

$$+ \frac{M \log \frac{6}{\delta}}{n} + 16 \times 2^{\frac{3}{4}} \sqrt{e}\beta \lceil \log_2 n \rceil \log(3e/\delta) + 32\sqrt{e}\beta \lceil \log_2 n \rceil \sqrt{\log 3e/\delta}$$

$$\leq 2\gamma \|A(S) - \mathbf{w}^*\|_2 \sqrt{\frac{\log \frac{6}{\delta}}{n}} + \sqrt{\frac{4\mathbb{E}_Z[\|\nabla f(\mathbf{w}^*; Z)\|_2^2] \log \frac{6}{\delta}}{n}} + \sqrt{\frac{\left(\frac{1}{2}\beta^2 + 32n\beta^2 \log(3/\delta)\right) \log \frac{6}{\delta}}{n}}$$

$$+ \frac{M \log \frac{6}{\delta}}{n} + 16 \times 2^{\frac{3}{4}} \sqrt{e}\beta \lceil \log_2 n \rceil \log(3e/\delta) + 32\sqrt{e}\beta \lceil \log_2 n \rceil \sqrt{\log 3e/\delta},$$

When $F(\mathbf{w})$ satisfies the PL condition and $\mathbf{w}^*$ is the projection of $A(S)$ onto the solution set $\arg\min_{\mathbf{w} \in \mathcal{W}} F(\mathbf{w})$, there holds the following error bound property (refer to Theorem 2 in Karimi et al. (2016))

$$\|\nabla F(A(S))\|_2 \geq \mu \|A(S) - \mathbf{w}^*\|_2.$$

Thus, we have

$$\mu \|A(S) - \mathbf{w}^*\|_2 \leq \|\nabla F(A(S))\|_2$$

$$\leq \|\nabla F_S(A(S))\|_2 + 2\gamma \|A(S) - \mathbf{w}^*\|_2 \sqrt{\frac{\log \frac{6}{\delta}}{n}} + \sqrt{\frac{4\mathbb{E}_Z[\|\nabla f(\mathbf{w}^*; Z)\|_2^2] \log \frac{6}{\delta}}{n}}$$

$$+ \sqrt{\frac{\left(\frac{1}{2}\beta^2 + 32n\beta^2 \log(3/\delta)\right) \log \frac{6}{\delta}}{n}} + \frac{M \log \frac{6}{\delta}}{n}$$

$$+ 16 \times 2^{\frac{3}{4}} \sqrt{e}\beta \lceil \log_2 n \rceil \log(3e/\delta) + 32\sqrt{e}\beta \lceil \log_2 n \rceil \sqrt{\log 3e/\delta}.$$

When $n \geq \frac{16\gamma^2 \log \frac{6}{\delta}}{\mu^2}$, we have $2\gamma \sqrt{\frac{\log \frac{6}{\delta}}{n}} \leq \frac{\mu}{2}$, then we can derive that

$$\mu \|A(S) - \mathbf{w}^*\|_2 \leq \|\nabla F(A(S))\|_2$$

$$\leq \|\nabla F_S(A(S))\|_2 + \frac{\mu}{2} \|A(S) - \mathbf{w}^*\|_2 + \sqrt{\frac{4\mathbb{E}_Z[\|\nabla f(\mathbf{w}^*; Z)\|_2^2] \log \frac{6}{\delta}}{n}}$$

$$+ \sqrt{\frac{\left(\frac{1}{2}\beta^2 + 32n\beta^2 \log(3/\delta)\right) \log \frac{6}{\delta}}{n}} + \frac{M \log \frac{6}{\delta}}{n}$$

$$+ 16 \times 2^{\frac{3}{4}} \sqrt{e}\beta \lceil \log_2 n \rceil \log (3e/\delta) + 32\sqrt{e}\beta \lceil \log_2 n \rceil \sqrt{\log 3e/\delta}.$$

This implies that

$$\|A(S) - \mathbf{w}^*\|_2$$

$$\leq \frac{2}{\mu} \Big( \|\nabla F_S(A(S))\|_2 + \sqrt{\frac{4\mathbb{E}_Z[\|\nabla f(\mathbf{w}^*; Z)\|_2^2] \log \frac{6}{\delta}}{n}}$$

$$+ \sqrt{\frac{\left(\frac{1}{2}\beta^2 + 32n\beta^2 \log(3/\delta)\right) \log \frac{6}{\delta}}{n}} + \frac{M \log \frac{6}{\delta}}{n} \tag{30}$$

$$+ 16 \times 2^{\frac{3}{4}} \sqrt{e}\beta \lceil \log_2 n \rceil \log (3e/\delta) + 32\sqrt{e}\beta \lceil \log_2 n \rceil \sqrt{\log 3e/\delta} \Big).$$

Then, substituting (30) into (29), when $n \geq \frac{16\gamma^2 \log \frac{6}{\delta}}{\mu^2}$, with probability at least $1 - \delta$

$$\|\nabla F(A(S)) - \nabla F_S(A(S))\|$$

$$\leq \|\nabla F_S(A(S))\| + 4\sqrt{\frac{\mathbb{E}_Z[\|\nabla f(\mathbf{w}^*; Z)\|^2] \log \frac{6}{\delta}}{n}} + 2\sqrt{\frac{\left(\frac{1}{2}\beta^2 + 32n\beta^2 \log(3/\delta)\right) \log \frac{6}{\delta}}{n}}$$

$$+ \frac{2M \log \frac{6}{\delta}}{n} + 32 \times 2^{\frac{3}{4}} \sqrt{e}\beta \lceil \log_2 n \rceil \log (3e/\delta) + 64\sqrt{e}\beta \lceil \log_2 n \rceil \sqrt{\log 3e/\delta}$$

$$\leq \|\nabla F_S(A(S))\| + C \left( \sqrt{\frac{2\mathbb{E}_Z[\|\nabla f(\mathbf{w}^*; Z)\|^2] \log \frac{6}{\delta}}{n}} + \frac{M \log \frac{6}{\delta}}{n} + e\beta \lceil \log_2 n \rceil \log (3e/\delta) \right), \tag{31}$$

where $C$ is a positive constant.

Since $F$ satisfies the PL condition with $\mu$, we have

$$F(A(S)) - F(\mathbf{w}^*) \leq \frac{\|\nabla F(A(S))\|^2}{2\mu}, \quad \forall \mathbf{w} \in \mathcal{W}. \tag{32}$$

So to bound $F(A(S)) - F(\mathbf{w}^*)$, we need to bound the term $\|\nabla F(A(S))\|^2$. And there holds

$$\|\nabla F(A(S))\|_2^2 = 2 \|\nabla F(A(S)) - \nabla F_S(A(S))\|^2 + 2\|\nabla F_S(A(S))\|_2^2. \tag{33}$$

On the other hand, when $f$ is nonnegative and $\gamma$-smooth, from Lemma 4.1 of Srebro et al. (2010), we have

$$\|\nabla f(\mathbf{w}^*; z)\|_2^2 \leq 4\gamma f(\mathbf{w}^*; z),$$

which implies that

$$\mathbb{E}_Z[\|\nabla f(\mathbf{w}^*; Z)\|_2^2] \leq 4\gamma \mathbb{E}_Z f(\mathbf{w}^*; Z) = 4\gamma F(\mathbf{w}^*). \tag{34}$$

Combing (31),(32), (33) and (34), using Cauchy-Bunyakovsky-Schwarz inequality, we can derive that

$$F(A(S)) - F(\mathbf{w}^*) \lesssim \|\nabla F_S(A(S))\|_2^2 + \frac{F(\mathbf{w}^*) \log (1/\delta)}{n} + \frac{\log^2(1/\delta)}{n^2} + \beta^2 \log^2 n \log^2(1/\delta).$$

The proof is complete.

$\square$

## D   PROOFS OF ERM

*Proof of Lemma 1.* Since $F_{S^{(i)}}(\mathbf{w}) = \frac{1}{n}\left(f(\mathbf{w}; z_i') + \sum_{j \neq i} f(\mathbf{w}, z_j)\right)$, we have

$$
\begin{aligned}
&F_S(\hat{\mathbf{w}}^*(S^{(i)})) - F_S(\hat{\mathbf{w}}^*(S)) \\
&= \frac{f(\hat{\mathbf{w}}^*(S^{(i)}); z_i) - f(\hat{\mathbf{w}}^*(S); z_i)}{n} + \frac{\sum_{j \neq i}(f(\hat{\mathbf{w}}^*(S^{(i)}); z_j) - f(\hat{\mathbf{w}}^*(S); z_j))}{n} \\
&= \frac{f(\hat{\mathbf{w}}^*(S^{(i)}); z_i) - f(\hat{\mathbf{w}}^*(S); z_i)}{n} + \frac{f(\hat{\mathbf{w}}^*(S); z_i') - f(\hat{\mathbf{w}}^*(S^{(i)}); z_i')}{n} \\
&\quad + \left(F_{S^{(i)}}(\hat{\mathbf{w}}^*(S^{(i)})) - F_{S^{(i)}}(\hat{\mathbf{w}}^*(S))\right) \\
&\leq \frac{f(\hat{\mathbf{w}}^*(S^{(i)}); z_i) - f(\hat{\mathbf{w}}^*(S); z_i)}{n} + \frac{f(\hat{\mathbf{w}}^*(S); z_i') - f(\hat{\mathbf{w}}^*(S^{(i)}); z_i')}{n} \\
&\leq \frac{2M}{n}\|\hat{\mathbf{w}}^*(S^{(i)}) - \hat{\mathbf{w}}^*(S)\|_2,
\end{aligned}
$$

where the first inequality follows from the fact that $\hat{\mathbf{w}}^*(S^{(i)})$ is the ERM of $F_{S^{(i)}}$ and the second inequality follows from the Lipschitz property. Furthermore, for $\hat{\mathbf{w}}^*(S^{(i)})$, the convexity of $f$ and the strongly-convex property of $F_S$ imply that its closest optima point of $F_S$ is $\hat{\mathbf{w}}^*(S)$ (the global minimizer of $F_S$ is unique). Then, there holds that

$$
F_S(\hat{\mathbf{w}}^*(S^{(i)})) - F_S(\hat{\mathbf{w}}^*(S)) \geq \frac{\mu}{2}\|\hat{\mathbf{w}}^*(S^{(i)}) - \hat{\mathbf{w}}^*(S)\|_2^2.
$$

Then we get

$$
\frac{\mu}{2}\|\hat{\mathbf{w}}^*(S^{(i)}) - \hat{\mathbf{w}}^*(S)\|_2^2 \leq F_S(\hat{\mathbf{w}}^*(S^{(i)})) - F_S(\hat{\mathbf{w}}^*(S)) \leq \frac{2M}{n}\|\hat{\mathbf{w}}^*(S^{(i)}) - \hat{\mathbf{w}}^*(S)\|_2,
$$

which implies that $\|\hat{\mathbf{w}}^*(S^{(i)}) - \hat{\mathbf{w}}^*(S)\|_2 \leq \frac{4M}{n\mu}$. Combined with the smoothness property of $f$ we obtain that for any $S^{(i)}$ and $S$

$$
\forall z \in \mathcal{Z}, \quad \left\|\nabla f(\hat{\mathbf{w}}^*(S^{(i)}); z) - \nabla f(\hat{\mathbf{w}}^*(S); z)\right\|_2 \leq \frac{4M\gamma}{n\mu}.
$$

The proof is complete. □

*Proof of Theorem 5.* From Lemma 1, we have $\|\nabla f(\hat{\mathbf{w}}^*(S); z) - \nabla f(\hat{\mathbf{w}}^*(S'); z)\|_2 \leq \frac{4M\gamma}{n\mu}$. Since $\nabla F_S(\hat{\mathbf{w}}^*) = 0$, we have $\|\nabla F_S(\hat{\mathbf{w}}^*)\|_2 = 0$. According to Theorem 4, we can derive that

$$
F(\hat{\mathbf{w}}^*(S)) - F(\mathbf{w}^*) \lesssim \frac{F(\mathbf{w}^*)\log(1/\delta)}{n} + \frac{\log^2 n \log^2(1/\delta)}{n^2}.
$$

□

## E   PROOFS OF PGD

*Proof of Theorem 6.* According to smoothness assumption and $\eta = 1/\gamma$, we can derive that

$$
\begin{aligned}
&F_S(\mathbf{w}_{t+1}) - F_S(\mathbf{w}_t) \\
&\leq \langle \mathbf{w}_{t+1} - \mathbf{w}_t, \nabla F_S(\mathbf{w}_t)\rangle + \frac{\gamma}{2}\|\mathbf{w}_{t+1} - \mathbf{w}_t\|_2^2 \\
&= -\eta_t\|\nabla F_S(\mathbf{w}_t)\|_2^2 + \frac{\gamma}{2}\eta_t^2\|\nabla F_S(\mathbf{w}_t)\|_2^2 \\
&= \left(\frac{\gamma}{2}\eta_t^2 - \eta_t\right)\|\nabla F_S(\mathbf{w}_t)\|_2^2 \\
&\leq -\frac{1}{2}\eta_t\|\nabla F_S(\mathbf{w}_t)\|_2^2.
\end{aligned}
$$

According to above inequality and the assumptions that $F_S$ is $\mu$-strongly convex, we can prove that

$$F_S(\mathbf{w}_{t+1}) - F_S(\mathbf{w}_t) \leq -\frac{1}{2}\eta_t\|\nabla F_S(\mathbf{w}_t)\|_2^2 \leq -\mu\eta_t(F_S(\mathbf{w}_t) - F_S(\hat{\mathbf{w}}^*)),$$

which implies that

$$F_S(\mathbf{w}_{t+1}) - F_S(\hat{\mathbf{w}}^*) \leq (1 - \mu\eta_t)(F_S(\mathbf{w}_t) - F_S(\hat{\mathbf{w}}^*)).$$

According to the property for $\gamma$-smooth for $F_S$ and the property for $\mu$-strongly convex for $F_S$, we have

$$\frac{1}{2\gamma}\|\nabla F_S(\mathbf{w})\|_2^2 \leq F_S(\mathbf{w}) - F_S(\hat{\mathbf{w}}^*) \leq \frac{1}{2\mu}\|\nabla F_S(\mathbf{w})\|_2^2,$$

which means that $\frac{\mu}{\gamma} \leq 1$.

Then If $\eta_t = 1/\gamma$, $0 \leq 1 - \mu\eta_t < 1$, taking over $T$ iterations, we get

$$F_S(\mathbf{w}_{t+1}) - F_S(\hat{\mathbf{w}}^*) \leq (1 - \mu\eta_t)^T(F_S(\mathbf{w}_t) - F_S(\hat{\mathbf{w}}^*)). \tag{35}$$

Combined (35), the smoothness of $F_S$ and the nonnegative property of $f$, it can be derive that

$$\|\nabla F_S(\mathbf{w}_{T+1}))\|_2^2 = O\left((1 - \frac{\mu}{\gamma})^T\right).$$

On the other hand, from Lemma 2, we have $\beta = \|\nabla f(\mathbf{w}_{T+1}(S); z) - \nabla f(\mathbf{w}_{T+1}(S'); z)\|_2 \leq \frac{2M\gamma}{n\mu}$. Since $\|\nabla F_S(\mathbf{w}_{T+1})\|_2 = O\left((1 - \frac{\mu}{\gamma})^T\right)$. According to Theorem 4, we can derive that

$$F(\mathbf{w}_{T+1}) - F(\mathbf{w}^*) \lesssim \left(1 - \frac{\mu}{\gamma}\right)^{2T} + \frac{F(\mathbf{w}^*)\log(1/\delta)}{n} + \frac{\log^2 n \log^2(1/\delta)}{n^2}.$$

Let $T \asymp \log n$, we have

$$F(\mathbf{w}_{T+1}) - F(\mathbf{w}^*) \lesssim \frac{F(\mathbf{w}^*)\log(1/\delta)}{n} + \frac{\log^2 n \log^2(1/\delta)}{n^2}.$$

The proof is complete. $\qquad\square$

## F  PROOFS OF SGD

We first introduce some necessary lemmata on the empirical risk.

**Lemma 12** ((Lei & Tang, 2021)). *Let $\{\mathbf{w}_t\}_t$ be the sequence produced by SGD with $\eta_t \leq \frac{1}{2\gamma}$ for all $t \in \mathbb{N}$. Suppose Assumption 1 hold. Assume for all $z$, the function $\mathbf{w} \mapsto f(\mathbf{w}; z)$ is $M$-Lipschitz and $\gamma$-smooth. Then, for any $\delta \in (0, 1)$, with probability at least $1 - \delta$, there holds that*

$$\sum_{k=1}^{t} \eta_k\|\nabla F_S(\mathbf{w}_k)\|_2^2 = O\left(\log\frac{1}{\delta} + \sum_{k=1}^{t}\eta_k^2\right).$$

**Lemma 13** ((Lei & Tang, 2021)). *Let $\{\mathbf{w}_t\}_t$ be the sequence produced by SGD with $\eta_t = \frac{2}{\mu(t+t_0)}$ such that $t_0 \geq \max\{\frac{4\gamma}{\mu}, 1\}$ for all $t \in \mathbb{N}$. Suppose Assumption 1 hold. Assume for all $z$, the function $\mathbf{w} \mapsto f(\mathbf{w}; z)$ is $M$-Lipschitz and $\gamma$-smooth and assume $F_S$ satisfies PL condition with parameter $\mu$. Then, for any $\delta \in (0, 1)$, with probability at least $1 - \delta$, there holds that*

$$F_S(\mathbf{w}_{T+1}) - F_S(\hat{\mathbf{w}}^*) = O\left(\frac{\log(T)\log^3(1/\delta)}{T}\right).$$

**Lemma 14** ((Lei & Tang, 2021)). *Let $e$ be the base of the natural logarithm. There holds the following elementary inequalities.*

- *If $\theta \in (0, 1)$, then $\sum_{k=1}^{t} k^{-\theta} \leq t^{1-\theta}/(1 - \theta)$;*

- *If $\theta = 1$, then $\sum_{k=1}^{t} k^{-\theta} \leq \log(et)$;*

- *If $\theta > 1$, then $\sum_{k=1}^{t} k^{-\theta} \leq \frac{\theta}{\theta-1}$.*

*Proof of Lemma 3.* We have known that $F_{S^{(i)}}(\mathbf{w}) = \frac{1}{n}\left(f(\mathbf{w}; z_i') + \sum_{j \neq i} f(\mathbf{w}; z_j)\right)$. We denote $\hat{\mathbf{w}}^*(S^{(i)})$ be the ERM of $F_{S^{(i)}}(\mathbf{w})$ and $\hat{\mathbf{w}}_S^*$ be the ERM of $F_S(\mathbf{w})$. From Lemma 1, we know that

$$\forall z \in \mathcal{Z}, \quad \left\|\nabla f(\hat{\mathbf{w}}^*(S^{(i)}); z) - f(\hat{\mathbf{w}}^*(S); z)\right\|_2 \leq \frac{4M\gamma}{n\mu}.$$

Also, for $\mathbf{w}_t$, the convexity of $f$ and the strongly-convex property implies that its closest optima point of $F_S$ is $\hat{\mathbf{w}}^*(S)$ (the global minimizer of $F_S$ is unique). Then, there holds that

$$\frac{\mu}{2}\|\mathbf{w}_t - \hat{\mathbf{w}}^*(S)\|_2^2 \leq F_S(\mathbf{w}_t) - F_S(\hat{\mathbf{w}}^*(S)) = \epsilon_{opt}(\mathbf{w}_t).$$

Thus we have $\|\mathbf{w}_t - \hat{\mathbf{w}}^*(S)\|_2 \leq \sqrt{\frac{2\epsilon_{opt}(\mathbf{w}_t)}{\mu}}$. A similar relation holds between $\hat{\mathbf{w}}^*(S^{(i)})$ and $\mathbf{w}_t^i$. Combined with the Lipschitz property of $f$ we obtain that for $\forall z \in \mathcal{Z}$, there holds that

$$\left\|\nabla f(\mathbf{w}_t; z) - \nabla f(\mathbf{w}_t^i; z)\right\|_2$$

$$\leq \|\nabla f(\mathbf{w}_t; z) - \nabla f(\hat{\mathbf{w}}^*(S); z)\|_2 + \left\|\nabla f(\hat{\mathbf{w}}^*(S); z) - \nabla f(\hat{\mathbf{w}}^*(S^{(i)}); z)\right\|_2$$

$$+ \left\|\nabla f(\hat{\mathbf{w}}^*(S^{(i)}); z) - \nabla f(\mathbf{w}_t^i; z)\right\|_2$$

$$\leq \gamma\|\mathbf{w}_t - \hat{\mathbf{w}}^*(S)\|_2 + \frac{4M\gamma}{n\mu} + \gamma\|\hat{\mathbf{w}}^*(S^{(i)}) - \mathbf{w}_t^i\|_2$$

$$\leq \gamma\sqrt{\frac{2\epsilon_{opt}(\mathbf{w}_t)}{\mu}} + \frac{4M\gamma}{n\mu} + \gamma\sqrt{\frac{2\epsilon_{opt}(\mathbf{w}_t^i)}{\mu}}.$$

According to Lemma 13, for any dataset $S$, the optimization error $\epsilon_{opt}(\mathbf{w}_t)$ is uniformly bounded by the same upper bound. Therefore, we write $\left\|\nabla f(\mathbf{w}_t; z) - \nabla f(\mathbf{w}_t^i; z)\right\|_2 \leq 2\gamma\sqrt{\frac{2\epsilon_{opt}(\mathbf{w}_t)}{\mu}} + \frac{4M\gamma}{n\mu}$ here.

The proof is complete. $\square$

Now We begin to prove Lemma 7.

*Proof of Lemma 7.* If $f$ is $L$-Lipschitz and $\gamma$-smooth and $F_S$ is $\mu$-strongly convex, according to (31) in the proof of Theorem 4, we know that for all $\mathbf{w} \in \mathcal{W}$ and any $\delta \in (0, 1)$, with probability at least $1 - \delta/2$, when $n > \frac{16\gamma^2 \log \frac{6}{\delta}}{\mu^2}$, we have

$$\left(\sum_{t=1}^{T} \eta_t\right)^{-1} \sum_{t=1}^{T} \eta_t \|\nabla F(\mathbf{w}_t)\|_2^2$$

$$\leq 16\left(\sum_{t=1}^{T} \eta_t\right)^{-1} \sum_{t=1}^{T} \eta_t \|\nabla F_S(\mathbf{w}_t)\|_2^2 + \frac{4C^2 L^2 \log^2 \frac{6}{\delta}}{n^2} + \frac{8C^2 \mathbb{E}_Z[\|\nabla f(\mathbf{w}^*; Z)\|_2^2] \log^2 \frac{6}{\delta}}{n} \quad (36)$$

$$+ \left(\sum_{t=1}^{T} \eta_t\right)^{-1} \sum_{t=1}^{T} \eta_t C^2 e^2 \beta_t^2 \lceil \log_2 n \rceil^2 \log^2 (3e/\delta),$$

where $\beta_t = \left\|\nabla f(\mathbf{w}_t; z) - \nabla f(\mathbf{w}_t^i; z)\right\|_2$ and $C$ is a positive constant.

From Lemma 3, we have $\left\| \nabla f(\mathbf{w}_t; z) - \nabla f(\mathbf{w}_t^i; z) \right\|_2 \leq 2\gamma \sqrt{\frac{2\epsilon_{opt}(\mathbf{w}_t)}{\mu}} + \frac{4M\gamma}{n\mu}$, thus

$$
\begin{aligned}
\beta_t^2 &= \left\| \nabla f(\mathbf{w}_t; z) - \nabla f(\mathbf{w}_t^i; z) \right\|_2^2 \\
&\leq \left( 2\gamma \sqrt{\frac{2\epsilon_{opt}(\mathbf{w}_t)}{\mu}} + \frac{4M\gamma}{n\mu} \right)^2 \\
&\leq \frac{16\gamma^2 (F_S(\mathbf{w}_t) - F_S(\hat{\mathbf{w}}^*(S)))}{\mu} + \frac{32M^2\gamma^2}{n^2\mu^2} \\
&\leq \frac{8\gamma^2 \|\nabla F_S(\mathbf{w}_t)\|_2^2}{\mu^2} + \frac{32M^2\gamma^2}{n^2\mu^2},
\end{aligned}
\tag{37}
$$

where the second inequality holds from Cauchy-Bunyakovsky-Schwarz inequality and the second inequality satisfies because $F_S$ is $\mu$-strongly convex.

Plugging (37) into (36), with probability at least $1 - \delta/2$, when $n > \frac{16\gamma^2 \log \frac{6}{\delta}}{\mu^2}$, we have

$$
\left( \sum_{t=1}^{T} \eta_t \right)^{-1} \sum_{t=1}^{T} \eta_t \|\nabla F(\mathbf{w}_t)\|_2^2
$$

$$
\leq \left( 16 + \frac{8\gamma^2 C^2 e^2 \lceil \log_2 n \rceil^2 \log^2 (6e/\delta)}{\mu^2} \right) \left( \sum_{t=1}^{T} \eta_t \right)^{-1} \sum_{t=1}^{T} \eta_t \|\nabla F_S(\mathbf{w}_t)\|_2^2
$$

$$
+ \frac{4C^2 L^2 \log^2 \frac{12}{\delta}}{n^2} + \frac{8C^2 \mathbb{E}_Z[\|\nabla f(\mathbf{w}^*; Z)\|_2^2] \log^2 \frac{12}{\delta}}{n} + \frac{32L^2\gamma^2 C^2 e^2 \lceil \log_2 n \rceil^2 \log^2 (6e/\delta)}{n^2\mu^2},
\tag{38}
$$

When $\eta_t = \eta_1 t^{-\theta}$, $\theta \in (0, 1)$, with $\eta_1 \leq \frac{1}{2\beta}$ and Assumption 1, according to Lemma 12 and Lemma 14, we obtain the following inequality with probability at least $1 - \delta/2$,

$$
\left( \sum_{t=1}^{T} \eta_t \right)^{-1} \sum_{t=1}^{T} \eta_t \|\nabla F_S(\mathbf{w}_t)\|^2 = \begin{cases} O\left( \frac{\log(1/\delta)}{T^{-\theta}} \right), & \text{if } \theta < 1/2 \\ O\left( \frac{\log(1/\delta)}{T^{-\frac{1}{2}}} \right), & \text{if } \theta = 1/2 \\ O\left( \frac{\log(1/\delta)}{T^{\theta-1}} \right), & \text{if } \theta > 1/2. \end{cases}
\tag{39}
$$

On the other hand, when $f$ is nonegative and $\gamma$-smooth, from Lemma 4.1 of Srebro et al. (2010), we have

$$
\|\nabla f(\mathbf{w}^*; z)\|_2^2 \leq 4\gamma f(\mathbf{w}^*; z),
$$

which implies that

$$
\mathbb{E}_Z[\|\nabla f(\mathbf{w}^*; Z)\|_2^2] \leq 4\gamma \mathbb{E}_Z f(\mathbf{w}^*; Z) = 4\gamma F(\mathbf{w}^*).
\tag{40}
$$

Plugging (40), (39) into (38), with probability at least $1 - \delta$, we derive that

$$
\left( \sum_{t=1}^{T} \eta_t \right)^{-1} \sum_{t=1}^{T} \eta_t \|\nabla F(\mathbf{w}_t)\|_2^2
$$

$$
= \begin{cases} O\left( \frac{\log^2 n \log^3(1/\delta)}{T^{-\theta}} \right) + O\left( \frac{\log^2 n \log^2(1/\delta)}{n^2} + \frac{F(\mathbf{w}^*) \log^2(1/\delta)}{n} \right), & \text{if } \theta < 1/2 \\ O\left( \frac{\log^2 n \log^3(1/\delta)}{T^{-\frac{1}{2}}} \right) + O\left( \frac{\log^2 n \log^2(1/\delta)}{n^2} + \frac{F(\mathbf{w}^*) \log^2(1/\delta)}{n} \right), & \text{if } \theta = 1/2 \\ O\left( \frac{\log^2 n \log^3(1/\delta)}{T^{\theta-1}} \right) + O\left( \frac{\log^2 n \log^2(1/\delta)}{n^2} + \frac{F(\mathbf{w}^*) \log^2(1/\delta)}{n} \right), & \text{if } \theta > 1/2. \end{cases}
$$

When $\theta < 1/2$, we set $T \asymp n^{\frac{2}{\theta}}$ and assume $F(\mathbf{w}^*) = O(\frac{1}{n})$, then we obtain the following result with probability at least $1 - \delta$

$$
\left( \sum_{t=1}^{T} \eta_t \right)^{-1} \sum_{t=1}^{T} \eta_t \|\nabla F(\mathbf{w}_t)\|_2^2 = O\left( \frac{\log^2 n \log^3(1/\delta)}{n^2} \right).
$$

When $\theta = 1/2$, we set $T \asymp n^4$ and assume $F(\mathbf{w}^*) = O(\frac{1}{n})$, then we obtain the following result with probability at least $1 - \delta$

$$\left(\sum_{t=1}^{T} \eta_t\right)^{-1} \sum_{t=1}^{T} \eta_t \|\nabla F(\mathbf{w}_t)\|_2^2 = O\left(\frac{\log^2 n \log^3(1/\delta)}{n^2}\right).$$

When $\theta > 1/2$, we set $T \asymp n^{\frac{2}{1-\theta}}$ and assume $F(\mathbf{w}^*) = O(\frac{1}{n})$, then we obtain the following result with probability at least $1 - \delta$

$$\left(\sum_{t=1}^{T} \eta_t\right)^{-1} \sum_{t=1}^{T} \eta_t \|\nabla F(\mathbf{w}_t)\|_2^2 = O\left(\frac{\log^2 n \log^3(1/\delta)}{n^2}\right).$$

The proof is complete.

$\square$

*Proof of Theorem 8.* Since $F$ is $\mu$-strongly convex, we have

$$F(\mathbf{w}) - F(\mathbf{w}^*) \leq \frac{\|\nabla F(\mathbf{w})\|_2^2}{2\mu}, \quad \forall \mathbf{w} \in \mathcal{W}. \tag{41}$$

So to bound $F(\mathbf{w}_{T+1}) - F(\mathbf{w}^*)$, we need to bound the term $\|\nabla F(\mathbf{w}_{T+1})\|_2^2$. And there holds

$$\|\nabla F(\mathbf{w}_{T+1})\|_2^2 = 2 \|\nabla F(\mathbf{w}_{T+1}) - \nabla F_S(\mathbf{w}_{T+1})\|^2 + 2\|\nabla F_S(\mathbf{w}_{T+1})\|_2^2. \tag{42}$$

From (31) in the proof of Theorem 4, if $f$ is $L$-Lipschitz and $\gamma$-smooth and $F_S$ is $\mu$-strongly convex, for all $\mathbf{w} \in \mathcal{W}$ and any $\delta > 0$, when $n \geq \frac{16\gamma^2 \log \frac{6}{\delta}}{\mu^2}$, with probability at least $1 - \delta/2$, there holds

$$\|\nabla F(\mathbf{w}_{T+1}) - \nabla F_S(\mathbf{w}_{T+1})\|_2$$

$$\leq \|\nabla F_S(\mathbf{w}_{T+1})\|_2 + C \left(\sqrt{\frac{2\mathbb{E}_Z[\|\nabla f(\mathbf{w}^*; Z)\|_2^2] \log \frac{12}{\delta}}{n}} + \frac{M \log \frac{12}{\delta}}{n} + e\beta \lceil \log_2 n \rceil \log(6e/\delta)\right)$$

$$\leq \|\nabla F_S(\mathbf{w}_{T+1})\|_2 + C \left(\sqrt{\frac{8\gamma F(\mathbf{w}^*) \log \frac{12}{\delta}}{n}} + \frac{M \log \frac{12}{\delta}}{n} + e\beta \lceil \log_2 n \rceil \log(6e/\delta)\right),$$

$$\tag{43}$$

where the last inequality follows from Lemma 4.1 of Srebro et al. (2010) when $f$ is nonnegative and $\gamma$-smooth (see (40)) and $C$ is a positive constant. Then we can derive that

$$\|\nabla F(\mathbf{w}_{T+1}) - \nabla F_S(\mathbf{w}_{T+1})\|_2^2$$

$$\leq 4\|\nabla F_S(\mathbf{w}_{T+1})\|_2^2 + \frac{32C^2\gamma F(\mathbf{w}^*) \log \frac{12}{\delta}}{n} + \frac{4M^2 C^2 \log^2 \frac{12}{\delta}}{n^2} + 4e^2 \beta_{T+1}^2 \lceil \log_2 n \rceil^2 \log^2(6e/\delta).$$

$$\tag{44}$$

From Lemma 3, we have $\left\|\nabla f(\mathbf{w}_t; z) - \nabla f(\mathbf{w}_t^i; z)\right\|_2 \leq 2\gamma\sqrt{\frac{2\epsilon_{opt}(\mathbf{w}_t)}{\mu}} + \frac{4M\gamma}{n\mu}$, thus

$$\beta_t^2 = \left\|\nabla f(\mathbf{w}_t; z) - \nabla f(\mathbf{w}_t^i; z)\right\|_2^2$$

$$\leq \left(2\gamma\sqrt{\frac{2\epsilon_{opt}(\mathbf{w}_t)}{\mu}} + \frac{4M\gamma}{n\mu}\right)^2$$

$$\leq \frac{16\gamma^2(F_S(\mathbf{w}_t) - F_S(\hat{\mathbf{w}}^*(S)))}{\mu} + \frac{32M^2\gamma^2}{n^2\mu^2}$$

$$\leq \frac{8\gamma^2\|\nabla F_S(\mathbf{w}_t)\|_2^2}{\mu^2} + \frac{32M^2\gamma^2}{n^2\mu^2},$$

$$\tag{45}$$

where the second inequality holds from Cauchy-Bunyakovsky-Schwarz inequality and the second inequality satisfies because $F_S$ is $\mu$-strongly convex.

Plugging (45) into (44), with probability at least $1 - \delta/2$, when , we have

$$
\|\nabla F(\mathbf{w}_{T+1}) - \nabla F_S(\mathbf{w}_{T+1})\|_2^2
$$

$$
\leq \left( 4 + 32e^2 \lceil \log_2 n \rceil^2 \log^2 (6e/\delta) \right) \|\nabla F_S(\mathbf{w}_{T+1})\|_2^2 + \frac{32C^2 \gamma F(\mathbf{w}^*) \log \frac{6}{\delta}}{n}
$$

$$
+ \frac{4L^2 C^2 \log^2 \frac{12}{\delta}}{n^2} + \frac{128M^2 \gamma^2 e^2 \lceil \log_2 n \rceil^2 \log^2 (6e/\delta)}{n^2 \mu^2}. \tag{46}
$$

According to the smoothness property of $F_S$ and Lemma 13, it can be derived that with propability at least $1 - \delta/2$

$$
\|\nabla F_S(\mathbf{w}_{T+1})\|_2^2 = O \left( \frac{\log T \log^3 (1/\delta)}{T} \right). \tag{47}
$$

Substituting (47), (46) into (42), we derive that

$$
\|\nabla F(\mathbf{w}_{T+1})\|_2^2
$$

$$
= O \left( \frac{\lceil \log_2 n \rceil^2 \log T \log^5 (1/\delta)}{T} \right) + O \left( \frac{\lceil \log_2 n \rceil^2 \log^2 (1/\delta)}{n^2} + \frac{F(\mathbf{w}^*) \log(1/\delta)}{n} \right). \tag{48}
$$

Further substituting (48) into (41), we have

$$
F(\mathbf{w}_{T+1}) - F(\mathbf{w}^*) = O \left( \frac{\lceil \log_2 n \rceil^2 \log T \log^5 (1/\delta)}{T} \right) + O \left( \frac{\lceil \log_2 n \rceil^2 \log^2 (1/\delta)}{n^2} + \frac{F(w^*) \log(1/\delta)}{n} \right).
$$

When choosing $T \asymp n^2$, we finally obtain that when $n$, with probability at least $1 - \delta$

$$
F(\mathbf{w}_{T+1}) - F(\mathbf{w}^*) = O \left( \frac{\log^4 n \log^5 (1/\delta)}{n^2} + \frac{F(\mathbf{w}^*) \log(1/\delta)}{n} \right).
$$

$\square$

# G   An Example for the 1-dimensional Mean Estimation Case

In this section, we provide an example for the 1-dimensional mean estimation case. If we denote the observed data as $X_1, X_2, \ldots, X_n$ drawn from an unknown distribution, we assume that this distribution is bounded (for example, any random variable $X$ that follows this distribution satisfies $a \leq X \leq b$).

To estimate the mean of the random variable, our objective function is defined as $f(\mu; X) = (X - \mu)^2$, which yields the usual least squares estimator. Using the definition of M-estimators, our goal becomes:

$$
\hat{\mu} = \arg \min_{\mu} \frac{1}{n} \sum_{i=1}^{n} (X_i - \mu)^2.
$$

We can easily verify that $f(\mu; X)$ satisfies the following properties: it is $4(b - a)$-Lipschitz, 2-strongly convex, and 2-smooth with respect to $\mu$. According to Theorem 5, we know that the mean estimated by this method, which minimizes the least squares loss (or the second central moment), will converge to

$$
\frac{\mathbb{V}[X] \log (1/\delta)}{n} + \frac{\log^2 n \log^2 (1/\delta)}{n^2}
$$

as the sample size $n$ increases, where $\mathbb{V}[X]$ is the variance of the distribution. It's worth noting that since this example pertains specifically to estimating the mean of a random variable, we have no additional parameters involved. Therefore, in this case, $F(w^*)$ in Theorem 5, which represents the variance of the distribution, is $O(1)$. Consequently, we cannot achieve a convergence rate of $1/n^2$. However, when our objective function employs more complex model functions, our method can achieve the faster rate of $O(1/n^2)$.

