# OpenReview forum: "Stability and Sharper Risk Bounds with Convergence Rate $O(1/n^2)$"
_ICLR.cc/2025/Conference — Submitted to ICLR 2025_

### Official Review · Reviewer_RDFx · 2024-10-28

**Soundness:** 2
**Presentation:** 3
**Contribution:** 2
**Rating:** 5
**Confidence:** 5

**Summary:**

The paper studied the algorithmic stability of the empirical risk minimization,  projected gradient descent, and SGD in the high probability form.
In particular, the authors show that  the rate $1/n^2$ can be achieved when the objective function is smooth and satisfied the PL condition (strongly convexity). The obtained results seems to the first of its kind in the literature.

**Strengths:**

1. The paper seems to be the first-ever-known fast rate in high probability in the framework of algorithmic stability.

2.  The paper is generally well written with good organization.

**Weaknesses:**

1.  While the results obtained are new and interesting, the proof techniques heavily reply on the existing literature (Klochkov & Zhivotovskiy, 2021) and specially Fan and Lei (2024). For instance, Theorem 1 is a refined version of Theorem 3 in Fan and lei (2024) by observing that the absolute constant  $M$ there can be replaced by the variance of the gradients of the loss.  The authors may need to further highlight  other technical novelty in the proof.

2.  The existing literature on the fast rates for SGD using stability approach often focused on the one-pass SGD, i.e., $T=n$. Although the fast rate was obtained in this paper for SGD with PL and smooth condition here,  the paper needs high gradient complexity, i.e. $T=n^2$.  In this case, there is a compromise here to achieve fast rate $1/n^2$ there.

**Questions:**

NA

---

> ### Author Response · Authors · 2024-11-21
> **Part I**
>
> Thank you for your review. Here are the replies for your questions.
>
> > Question 1: As mentioned in Weaknesses, while the results obtained are new and interesting, the proof techniques heavily reply on the existing literature (Klochkov \& Zhivotovskiy, 2021) and specially Fan and Lei (2024). For instance, Theorem 1 is a refined version of Theorem 3 in Fan and lei (2024) by observing that the absolute constant $M$ there can be replaced by the variance of the gradients of the loss. The authors may need to further highlight other technical novelty in the proof.
>
>
> Answer: Regarding the example you mentioned, it is true that, our result in Theorem 3 improves upon the conclusion of (Fan and Lei, 2024, Theorem 3) by showing that the absolute constant $ M $ can be replaced with the variance of the gradients of the loss. However, achieving this improvement is not straightforward. Inspired by (Klochkov \& Zhivotovskiy, 2021), but it is essential to note that prior work predominantly focuses on one-dimensional random variables. Our approach, however, confronted additional complexities due to the involvement of random vectors and their norms. In one-dimensional cases, the condition of a variable value being $0$ is equivalent to its absolute value being $0$. However, the Euclidean norm of a vector being $0$ does not imply that the vector itself is the $0$ vector, adding significant complexity to our proof. This distinction necessitated a more nuanced analysis during the proofs.
>
> Beside, for Theorem 1, in our writing, we discovered that the constant in Marcinkiewicz-Zygmund's inequality for random variables taking values in a Hilbert space could be further improved. Consequently, we included this part in the paper. Considering that Theorem 1 has other broad applications, we report this result as well. Although it is the constant-level improvement. The proof was challenging, as it involved establishing the best constant in Marcinkiewicz-Zygmund's inequality for random variables taking values in a Hilbert space, which has its foundations in Khintchine-Kahane's inequality. To prove the best constant, we utilized Stirling's formula for the Gamma function to construct appropriate functions, establishing both upper and lower bounds. Then using Mean Value Theorem, this approach ultimately led to the convergence of the constant as $p$ approaches infinity.
>
> For Theorem 4, the proof for this theorem primarily exploits the self-bounded properties of functions, facing challenges akin to those highlighted in Theorem 3, related to the interaction between the norms $ || \cdot || $ and the expectation symbol $ \mathbb{E} $. While we can sometimes apply the Jensen's inequality, there are instances where we must innovatively navigate around such relationships within the function construction. This complexity does not arise for scalar variables, and we think that this provide a pathway for future inquiries into the stability of algorithms involving vector norm considerations.
>
> In addition to the technical novelty mentioned above, as Reviewer 9oQf noted in weaknesses,
>
> > Given the impressive results, I do think this is a very minor weakness, but the technical novelty is indeed not that compelling. The main result is mostly derived from applying Klochkov and Zhivotovskiy's techniques to a more restrictive (in terms of problem well-behavedness constraints) and more general (in terms of studying the 'oracle' $ A $) setup. This might potentially bar the paper from entering the conference highlight.
>
> The novelty of our results are also important. We also highlight the novelty of our work for the results in public response. Notably, our results are asymptotically optimal, which align with existing theories. We not only provide a more precise stability analysis tool under PL conditions, but we also offer new and improved insights into the relationship between algorithm stability and generalization in non-convex settings (Theorem 3). This has potential applications in non-convex scenarios as well.

---

> > ### Author Response · Authors · 2024-11-21
> > **Part II**
> >
> > > Question 2: As mentioned in Weaknesses, the existing literature on the fast rates for SGD using stability approach often focused on the one-pass SGD, i.e., $T=n$. Although the fast rate was obtained in this paper for SGD with PL and smooth condition here, the paper needs high gradient complexity, i.e. $T=n^2$. In this case, there is a compromise here to achieve fast rate $1/n^2$ there.
> >
> >
> > Answer: Our results also demonstrate better performance on one-pass SGD comparing with existing work. To enhance the readability of this paper, we primarily focus on exploring the upper limits of generalization that can be achieved through stable algorithms. In fact, achieving the $ 1/n^2 $ bound is challenging and requires some relatively stringent assumptions. Here, we present our result on one-pass SGD. Our Theorem 8 yields an optimal upper bound
> > $$
> >  \text{excess risk}  = O \left(\frac{ \left\lceil \log_2 n \right\rceil^2 \log T \log^5(1/\delta)}{T}\right) + O \left(\frac{ \left\lceil \log_2 n \right\rceil^2 \log^2(1/\delta)}{n^2} + \frac{F(w^*) \log(1/\delta)}{n}\right).
> >     $$
> > According to our results, our bound is $O(1/n^2 + F(w^*)/n)$ under $T = n^2$.
> > If assumptions are $F(w^*) = O(1)$ and $T=n$, which are milder one-pass SGD assumptions, our result implies that the excess risk bound can be up to $1/n$, which is also the best high probability result. For comparison, there are two results in stability analysis that are similar to ours. One is the $ 1/n $ result for one-pass SGD (Lei \& Ying, 2020), but it pertains to the expected version. The high-probability version is significantly more challenging. Currently, the best result under the high-probability version is also $ 1/n $ (Li \& Liu, 2021), but (Li \& Liu, 2021) focuses on multi-pass SGD, which requires $ T = n^2 $.
> >
> >
> >
> > Reference
> >
> > Fan Jun and Yunwen Lei. High-probability generalization bounds for pointwise uniformly stable algorithms. Applied and Computational Harmonic Analysis.
> >
> > Yegor Klochkov, and Nikita Zhivotovskiy. Stability and deviation optimal risk bounds with convergence rate $O(1/n)$. NeurIPS 2021.
> >
> > Shaojie Li and Yong Liu. Improved learning rates for stochastic optimization: Two theoretical viewpoints. arXiv preprint arXiv:2107.08686, 2021.
> >
> > Yunwen Lei and Yiming Ying. Fine-grained analysis of stability and generalization for stochastic gradient descent. ICML 2020.

---

> > > ### Comment · Reviewer_RDFx · 2024-11-27
> > > **thanks**
> > >
> > > Thank you for the response.  I will keep my original score.

---

> ### Author Response · Authors · 2024-11-25
> **Follow-up on Rebuttal Submission**
>
> Dear Reviewer RDFx,
>
> We hope this message finds you well. We are writing to follow up on the rebuttal we submitted several days ago. We wanted to check in to see whether it has addressed your concerns.
>
> If there are any remaining questions or if there are new issues you would like to discuss, please feel free to reach out. We appreciate your feedback and am eager to ensure that all aspects of the paper meet your expectations.
> Thank you for your time and consideration.
>
>
> Best regards,
>
> Authors

---

### Official Review · Reviewer_XMnk · 2024-10-28

**Soundness:** 3
**Presentation:** 3
**Contribution:** 2
**Rating:** 5
**Confidence:** 3

**Summary:**

The paper presents high-probability generalization bounds based on stability analysis. To this aim, the paper first presents a moment bound for sums of vector-valued functions of independent variables. Based on this, the paper gives a high-probability bound on the difference between the gradients of population risks and the gradients of empirical risks. This further implies excess risk bounds under a PL condition. The paper shows that this risk bound can be of order of $O(1/n^2)$ under some conditions. Further applications to empirical risk minimization, gradient descent and stochastic gradient descent are also given.

**Strengths:**

- The paper shows improved generalization bounds based on stability analysis. The paper gives clear comparison with existing results. For example, Remark 4 and Remark 7 show the advantage of the derived generalization bounds over the existing results in Fan & Lei 2024 and Xu & Zeevi (2024).
- The paper also shows improved excess risk bounds over the existing results. For example, in Remark 9, the paper shows that the results outperform those in Zhang & Zhou by removing the dependency on the dimensionality. In Remark 11, the paper shows that it is better than Klochkov & Zhivotovskiy (2021) by allowing for rates of order $O(1/n^2)$.

**Weaknesses:**

- While the rates of order $O(1/n^2)$ are interesting, it seems that the bounds have another linear dependency on $1/\mu^2$. For example, according to proof of Theorem 4, there is a missing factor of $1/\mu^2$ in each term on the right-hand side of the bound. Note $\mu$ is the PL parameter and can be very small in practice. This can make the derived bound less interesting in practice.
- For applications in Section 4, the paper considers strongly convex problems. There is an assumption that $n\geq 16\gamma^2\log(6/\delta)/\mu^2$, which amounts to saying that $\mu \geq 4\gamma\log^{1/2}(6/\delta)/\sqrt{n}$. Note that typically one needs to introduce a strongly convex regularizer to get strong convexity. However, one is mostly interested in the original loss function without the regularizer. While the paper shows improved rates for the loss with the regularizer, the requirement $\mu \geq 4\gamma\log^{1/2}(6/\delta)/\sqrt{n}$ makes the rate for the loss without the regularizer still not fast.
- For SGD, in Remark 13 the paper requires $T=n^4$ to get good bounds. This requirement on the number of iterations is a bit large. Then, the computational cost may be huge in this case.

**Questions:**

- Can the results in Section 4 guarantee fast rates for the original loss without the regularizer? That is, whether we can transform the generalization bounds in Section 4 to get fast rates on the decay of the population risk without the regularizer?
- Lemma 3 is a bit strange. The left hand side involves w_t^i and w_t. However, the right-hand side only involves $\epsilon(w_t)$. The authors should check it.

Minor comments:
- "under strongly convexity" in Abstract
- $w^*$ is not defined in the introduction
- "inequality which provide" in Section 3

---

> ### Author Response · Authors · 2024-11-21
> **Part I**
>
> Thank you for your review. We want to emphasize that the advancements in the relationship between stability and generalization are limited recent years. Most papers focus on utilizing stability tools to address application problems or to explore the stability of specific algorithms. In contrast, our analysis of stability bounds dependent on optimal parameters is an improvement in this area. We also highlight the novelty of our results and the technical innovations in public response. Here are the replies for your questions.
>
>
> > Question 1: As mentioned in Weaknesses, while the rates of order $O(1/n^2)$ are interesting, it seems that the bounds have another linear dependency on $1/\mu^2$. For example, according to proof of Theorem 4, there is a missing factor of $1/\mu^2$ in each term on the right-hand side of the bound. Note $\mu$ is the PL parameter and can be very small in practice. This can make the derived bound less interesting in practice.
>
>
> Answer: Our Theorem 4 only requires that the population risk $ F $ satisfies the PL condition with parameter $ \mu $, which is relatively weak. In fact, $ \mu $ can be viewed as a constant in some cases within classical learning theory. For instance, consider the classical linear regression setup: $ f(w; (x,y)) = || y - \langle w, x \rangle ||^2 $, where $ y = \langle w^*, x \rangle + v $, with $ v \sim N(0, 1) $ and $ x \sim N(0, \sigma I_{d \times d}) $. In this case, we have $ \mu = 2 $.

---

> > ### Author Response · Authors · 2024-11-21
> > **Part II**
> >
> > > Question 2: As mentioned in Weaknesses, for applications in Section 4, the paper considers strongly convex problems. There is an assumption ...
> >
> > Answer: First, we would like to discuss the optimality of our results from the perspective of asymptotic optimality. Our results are asymptotically optimal, aligning with existing theories. For instance, in the case of ERM, according to the classical asymptotic theory, under some local regularity assumptions, when $ n \rightarrow \infty $, it is shown in the asymptotic statistics monographs (Vaart, 2000) that
> > $$
> >     \sqrt{n} (\hat{w}^*(S) - w^*) \stackrel{\rho}\longrightarrow N(0, H^{-1}Q H^{-1}),
> > $$
> > where $ \hat{w}^*(S) $ denotes the ERM algorithm, $ H = \nabla^2 F(w^*) $, $Q$ is the covariance matrix of the loss gradient at $w^*$ (also called Fisher's information matrix): $ Q = \mathbb{E} [\nabla f(w^*; z) \nabla f(w^*; z)^T ] $
> > ($A^T$ denotes the transpose of a matrix $A$), and $\stackrel{\rho}\longrightarrow$ means convergence in distribution. The second-order Tayplr expansion of the population risk around $w^*$ then allows to derive the same asymptotic law for the scaled excess risk $ 2n (F(\hat{w}^*(S)) - F(w^*)) $. Under suitable conditions, this asymptotic rate is usually theoretically optimal (Vaart, 1989). For example, when $f(w;z)$ is a negative log-likelihood, this asymptotic rate matches the Hajek-LeCam asymptotic minimax lower bound (Hajek, 1972; LeCam, 1972).
> >
> > We then analysis the result in Theorem 5. In the proof of Theorem 4, before we use the self-bounded smoothness property $ || \nabla f(w^*;z) ||^2 \leq 4 \gamma f(w^*;z) $, we get the following result for Theorem 5
> > $$
> >     F(\hat{w}^*(S)) - F(w^*) \lesssim \frac{ \mathbb{E}||\nabla f(w^*;z)||^2 \log(1/\delta)}{\mu n} + \frac{\log^2 (1/\delta)}{n^2}.
> > $$
> > Our result is the finite sample version of the asymptotic rate given above, which characterizes the cirtical sample size sufficient to enter this ``asymptotic regime''. This is because the excess risk error $F(\hat{w}^*(S)) - F(w^*)$ can be approximated by the quadratic form $ (\hat{w}^*(S) - w^*)^T H (\hat{w}^*(S) - w^*) $. $ 1/\mu $ is a natural proxy for the inverse Hessian $H^{-1}$, and $\mathbb{E}[ ||\nabla f(w^*;z)||^2 ]$ is a natural proxy for $Q$. Furthermore, when discussing sample complexity, (Xu \& Zeevi, 2024) constructed a simple linear model to demonstrate the constant-level optimality of the sample complexity lower bound $\Omega(d \beta^2/\mu^2)$ under such conditions. Our theorem further reveals, through the use of stability methods, that this complexity lower bound can be independent of the dimensionality $d$. Our results are consistent with existing statistical learning theory; under fast rates, this condition is unavoidable.
> >
> > The primary focus of this paper is to explore the extent to which stability tools can achieve optimal results. In fact, fast rates are not easily attainable; most papers discussing fast rates, such as (Klochkov \& Zhivotovskiy, 2021), impose very stringent assumptions. Building upon this foundation, we have achieved improved results under the same assumptions, while also corroborating the asymptotic optimal theory.
> >
> > Our work further advances the development of stability and generalization tools. As described in our public response, while the main theme of this paper is to discuss fast rates, our results also demonstrate improved performance under weaker conditions. Previous analyses based on (Klochkov \& Zhivotovskiy, 2021) were constrained by requirements for algorithm stability; when specific algorithm stability is $ \beta = O(1/\sqrt{n}) $, their analysis limits the upper bound to a rate of $ 1/\sqrt{n} $. In contrast, our results provide an upper bound analysis of up to $ 1/n $. Compared to other generalization tools, such as complexity methods, our analysis has also improved the sample complexity results by removing the dependence on the dimensionality $ d $.
> >
> > Finally, some of the theorems presented in this paper may offer a new perspective. Our Theorem 3 does not rely on strong convexity assumptions. Currently, (Li & Liu, 2021) have provided results regarding the high-probability stability of SGD under non-convex conditions, achieving a convergence rate of $ T^{c\gamma}\sqrt{1/n} $, where $ c $ is a constant. Clearly, this convergence rate falls short of the $ O(1/n^2) $ requirement established in our paper. In fact, if we directly apply this conclusion to our theorem regarding the relationship between gradients of generalization error bounds and algorithm stability (Theorem 3), we can derive a new result—potentially the best in the non-convex setting—with weakened conditions, which does not introduce additional complexity to the proof but contributes little to the overall discussion in our paper. However, we believe that Theorem 3 provides better tools for further exploration of generalization analysis in non-convex settings.

---

> > > ### Author Response · Authors · 2024-11-21
> > > **Part III**
> > >
> > > > Question 3: As mentioned in Weaknesses, For SGD, in Remark 13 the paper requires $T = n^4$ to get good bounds. This requirement on the number of iterations is a bit large. Then, the computational cost may be huge in this case.
> > >
> > > Answer: Indeed, since the main thrust of our paper relies on the $O(1/n^2)$ upper bound associated with optimal parameters, as you pointed out, Remark 13 indicates that when the learning rate $\eta$ decays as $\eta = \eta_0 t^{-1/2}$ with increasing iterations, the optimal upper bound for the gradients of population risk requires $T = n^4$. However, this does not imply that our results are inferior under other conditions. Regarding the setting you pointed out (Theorem 7), when we set $ T = n^2 $, we can still achieve an upper bound of $ O(1/n) $. This represents the best current result under high probability version. (Li & Liu, 2021) also provided a similar upper bound under the same conditions, but the parameter $n$ depends on the dimensionality $d$ of the parameters.
> > >
> > > Other theorems in Section 4 are similar. For instance, using PGD algorithm, the result in (Klochkov \& Zhivotovskiy, 2021) is
> > > $$
> > > \text{excess risk} = O\Big( \frac{\log(1/\delta)}{n} \Big).
> > > $$
> > > As a comparison, under the same assumptions (including the same iteration step $T = O(\log n)$), our results in Theorem 6 is
> > > $$
> > > \text{excess risk} = O\Big(\frac{\log^2(1/\delta)}{n^2} + \frac{ \log(1/\delta) F(w^*)}{n}\Big).
> > > $$
> > > As long as $ F(w^*) $ converges as $ n $ increases, our results outperform better. Even when $F(w^*) = O(1)$, our PGD results remain comparable to those presented by Klochkov \& Zhivotovskiy (2021).
> > >
> > >
> > > > Question 4: As mentioned in Questions, Can the results in Section 4 guarantee fast rates for the original loss without the regularizer? That is, whether we can transform the generalization bounds in Section 4 to get fast rates on the decay of the population risk without the regularizer?
> > >
> > >
> > > Answer: Please refer to the discussion in Question 2.
> > >
> > >
> > > > Question 5: As mentioned in Questions, Lemma 3 is a bit strange. The left hand side involves $w_t^i$ and $w_t$. However, the right-hand side only involves $\epsilon(w_t)$. The authors should check it.
> > >
> > >
> > > Answer: We have reviewed the proof of Lemma 3 and found no issues. In the proof section, we provided a brief explanation of why $\epsilon(w_t)$ and $\epsilon(w_t^{(i)})$ are equivalent. Here, we offer a more detailed explanation to help clarify this matter.
> > >
> > > $\epsilon(w_t)$ represents the optimization error of the model obtained after running the SGD algorithm for $t$ iterations on dataset $S$, compared to the optimal model on the same dataset. Similarly, $\epsilon(w_t^{(i)})$ denotes the optimization error after running the SGD algorithm for $t$ iterations on the dataset $S^{(i)}$. According to the optimization error results that we utilize later, both terms share a common upper bound. In other words, one can be used to substitute for the other due to their equivalence in terms of bounded behavior.
> > >
> > > > Question 6: As mentioned in Questions, minor comments...
> > >
> > > Answer: Thank you for your correction. We will try our best to fix these minor errors in the final version.
> > >
> > >
> > > Reference
> > >
> > >
> > > Yegor Klochkov, and Nikita Zhivotovskiy. Stability and deviation optimal risk bounds with convergence rate $O(1/n)$. NeurIPS 2021.
> > >
> > > Shaojie Li and Yong Liu. Improved learning rates for stochastic optimization: Two theoretical viewpoints. arXiv preprint arXiv:2107.08686, 2021.
> > >
> > > Yunbei Xu and Assaf Zeevi. Towards optimal problem dependent generalization error bounds in statastical learning theory. Mathematics of Operations Research 2024.

---

> ### Author Response · Authors · 2024-11-25
> **Follow-up on Rebuttal Submission**
>
> Dear Reviewer XMnk,
>
> We hope this message finds you well. We are writing to follow up on the rebuttal we submitted several days ago. We wanted to check in to see whether it has addressed your concerns.
>
> If there are any remaining questions or if there are new issues you would like to discuss, please feel free to reach out. We appreciate your feedback and am eager to ensure that all aspects of the paper meet your expectations.
> Thank you for your time and consideration.
>
>
> Best regards,
>
> Authors

---

> > ### Comment · Reviewer_XMnk · 2024-12-01
> > **Official Comment by Reviewer XMnk**
> >
> > Thank you for your response and the example to justify the parameter $\mu$ in Theorem 4. I think this example does not apply to the applications in Section 4. Indeed, results in Section 4 require $f$ to be $\mu$-strongly convex. Then an assumption on the distribution does not make $f$ to be strongly convex. Then, one needs to introduce a regularizer and then the regularization parameter $\mu$ is often small in practice.

---

> > > ### Author Response · Authors · 2024-12-02
> > > **Part I**
> > >
> > > Thank you for your comments. We appreciate your concerns and would like to address them based on the following four points:
> > >
> > > 1. The examples we provided in response to Question 1 can be applied to the applications in Section 4. Intuitively, as we stated in the first paragraph of Section 4, the introduction of the strongly-convex assumption is intended to establish the algorithm's uniform stability in gradients. It is crucial to note that in proving the algorithm's uniform stability in gradients, we need the inequalities $ F_S(w) - F_S(\hat{w}^*(S)) \geq \frac{\mu}{2} || w - \hat{w}^*(S) ||_2^2 $.
> > >
> > > To elaborate, consider the classical linear regression setup: $ f(w; (x,y)) = || y - \langle w, x \rangle ||^2 $, where $ y = \langle w^*, x \rangle + v $, with $ v \sim N(0, 1) $ and $ x \sim N(0, I_{d \times d}) $.
> > >
> > > For this linear regression example, we can easily derive that $\frac{\partial^2 f}{\partial w^2} = 2 x x^T$. it is straightforward to verify that the population risk $ F $ satisfies the PL condition with $ \mu = 2 $ because $x \sim N(0, I_{d \times d}) $ and $\mathbb{E} [x x^T] = I_{d \times d}$. Thus, we can use Theorem 4. For the empirical risk $F_S$, some adjustments are necessary in the proofs. Though we may not directly obtain the same condition for the empirical risk $ F_S $, notice that $ x \sim N(0, I_{d \times d}) $, the matrix $ \frac{1}{n}  \sum_{i=1}^n x_ix_i^T $ is a $ d \times d $ matrix with sum of $n$ independent rank-$1$ matrix.
> > > Then, $F_S$ satisfies PL condition for any $w \in \mathcal{W}$
> > > $$
> > > F_S(w) - F_S(\hat{w}^*(S)) \geq \left( \lambda_{\min}(\frac{1}{n}  \sum_{i=1}^n x_ix_i^T) \right) \left|| w - \hat{w}^*(S) \right||_2^2.
> > > $$
> > >
> > > According to Marchenko-Pastur low [Marchenko, 1967], with high probability, we have the following inequality $\lambda_{\min}(\frac{1}{n}  \sum_{i=1}^n x_ix_i^T) \geq 1 - \sqrt{\frac{d}{n}}$, which means that for any $w \in \mathcal{W},$ $F_S(w) - F_S(\hat{w}^*(S)) \geq \left( 1 -\sqrt{\frac{d}{n}} \right) || w - \hat{w}^*(S) ||_2^2$.
> > >
> > >
> > > In the remaining parts, we can follow the proof in Section 4, replacing the places where the strong convexity assumption was needed with the above inequality. By Boole's inequality (also known as the union bound in probability theory), we can derive similar conclusions. For instance, when considering ERM, we obtain a similar result:
> > >
> > > Let $ f $ satisfy the classical linear regression given above. Then, for any $ \delta \in (0,1) $, with probability at least $ 1 - \delta$, we have:
> > >
> > > $$
> > > F(\hat{w}^*(S)) - F(w^*) \lesssim \frac{F(w^*) \log{(1/\delta)}}{n} + \frac{ \log^2 n \log^2(1/\delta)}{n^2}.
> > > $$
> > >
> > > Similar conclusions can be drawn for PGD and SGD.
> > >
> > > 2. The following example of a one-dimensional mean estimation also satisfies our assumptions, and $\mu$ is not very small. To briefly outline the construction:
> > >
> > > Let the observed data be $ X_1, X_2, \ldots, X_n $ drawn from an unknown, bounded distribution (for instance, any random variable $ X $ that follows this distribution satisfies $ a \leq X \leq b $).
> > >
> > > To estimate the mean, our objective function is defined as $ f(m; X) = (X - m)^2 $, leading to the usual least squares estimator. Our goal becomes:
> > >
> > > $$
> > > \hat{m} = \arg\min_m \frac{1}{n} \sum_{i=1}^{n} (X_i - m)^2.
> > > $$
> > >
> > > We can easily verify that $ f(m; X) $ satisfies the properties of $ 4(b-a) $-Lipschitz, $ 2 $-strongly convex, and $ 2 $-smooth with respect to $ m $. According to Theorem 5, the mean estimated by this method will converge to:
> > >
> > > $$
> > > \frac{\mathbb{V}[X] \log{(1/\delta)}}{n} + \frac{\log^2 n \log^2(1/\delta)}{n^2}.
> > > $$
> > > as the sample size $n$ increases, where $\mathbb{V}[X]$ is the variance of the distribution. This classical statistical mean estimation scenario meets all our assumptions, with $ \mu $ not being overly small. We have added this example in the latest version in Appendix G.

---

> ### Author Response · Authors · 2024-12-02
> **Part II**
>
> 3. Regarding general machine learning setting, as you pointed out, regularization is often necessary. We provide a discussion concerning the hyperparameters of the regularizer. As detailed in the first point, the essence of the strong-convexity assumption is that $ F_S $ must satisfy a PL condition (or satisfy that PL condition with high probability). Considering the empirical risk with a regularizer, $ F_S(w) = \frac{1}{n} \sum_{i=1}^n f(w;z_i) + \frac{\mu}{2} ||w||_2^2 $, suppose $ \mu \asymp n^{-\alpha} $ with $ \alpha \in [0, \frac{1}{2}) $. Using PGD as an example, we substitute into the proof in Section 4 and derive the following result:
>
> $$
> \text{excess risk} = O\left( \frac{F(w^*)}{n^{1- \alpha}} + \frac{1}{n^{2- \alpha}} \right).
> $$
>
> This indicates that the sample complexity requirement becomes $ n^{1-2\alpha} = \Omega \left(16 \gamma^2 \log \left(\frac{6}{\delta}\right) \right)$, while our results can achieve $ O(1/n^{2 - \alpha}) $ when $ F(w^*) = O(1/n) $. Furthermore, when $ \alpha = 0 $, that is, when $ \mu $ is independent of $ n $, our results can tightly reach $ O(1/n^2) $. When $\alpha \in (0, \frac{1}{2})$, our results are also the best currently available. For comparison, the existing stability generalization error bounds $ R_S(A(S)) - R(A(S)) $ given by [Agarwal and Niyogi, 2009; Jin et al., 2009; Wang et al., 2019] are of the order $ O( 1/n^{\frac{1}{2} - \alpha} )$, while the excess risk bound  $ R(A(S)) - R(w^*) $ given by [Klochkov and Zhivotovskiy, 2021] is of the order $ O(M/n^{1 - \alpha}) $ under strongly convexity.
>
>
> 4. Finally, we want to emphasize that the core idea of this paper is to provide finer analytical tools for stability and generalization. As we discussed in the rebuttal and paper, our results perform better than existing ones under the same assumptions. **We have also discussed the optimality of our results in the latest version**. Our method can achieve rates as high as $ O(1/n^2) $ depending on the optimal parameters. As the saying goes, "no free lunch"; attaining such a fast rate is not simple and often necessitates some assumptions. Compared to other papers that achieve fast rates, our approach delivers tighter bounds without introducing more stringent assumptions.
> % We think it somewhat unreasonable to overly emphasize scenarios where our assumptions may not hold.
>
> We hope this clarifies our reasoning and addresses your concerns adequately. Thank you once again for your insights. If we haven't explained things clearly or if you have any questions, please feel free to reach out to us.
>
>
> Reference
>
> Marchenko, V. A.; Pastur, L. A. Distribution of eigenvalues for some sets of random matrices. in Mat. Sb. N.S. (in Russian) 1967.
>
> S. Agarwal and P. Niyogi. Generalization bounds for ranking algorithms via algorithmic stability. JMLR 2009.
>
>
> R. Jin, S. Wang, and Y. Zhou. Regularized distance metric learning: Theory and algorithm. NeurIPS, 2009.
>
> B. Wang, H. Zhang, P. Liu, Z. Shen, and J. Pineau. Multitask metric learning: Theory and algorithm. AISTATS, 2019.
>
> Y. Klochkov, and N. Zhivotovskiy. Stability and deviation optimal risk bounds with convergence rate $O(1/n)$. NeurIPS 2021.

---

### Official Review · Reviewer_9oQf · 2024-11-01

**Soundness:** 4
**Presentation:** 4
**Contribution:** 3
**Rating:** 8
**Confidence:** 3

**Summary:**

The authors provide dimension-free, $O(1/n^2)$-rate generalization bounds for learning algorithms with uniformly stable gradients. This is applicable to the ERMs of sufficiently well-behaved risk functions, as well as the gradient descent iterates of sufficiently well-behaved objective functions.

**Strengths:**

- As summarized in Table 1 (which is very informative), the bounds for ERM, PGD, and SGD are indeed the sharpest in the literature, in particular in terms of the sample complexity.

- Overall the paper is very clearly written, including proof sketch, comparison etc.

**Weaknesses:**

- Given the impressive results I do think this is a very minor weakness, but the technical novelty is indeed not *that* compelling. The main result is mostly derived from applying Klochkov and Zhivotovskiy's techniques to a more restrictive (in the sense of problem well-behavedness constraints) and more general (in the sense of studying the "oracle" $A$) set-up. This might potentially bar the paper from entering the conference highlight.

**Questions:**

- Can the authors briefly comment on the implications for 1-dimensional mean estimation (w/ empirical mean, M-estimators, etc.)? This might be a good illustrative example.

---

> ### Author Response · Authors · 2024-11-21
>
> Thank you for your review. Here are the replies for your questions.
>
> > Question 1: As mentioned in Weaknesses, given the impressive results I do think this is a very minor weakness, but the technical novelty is indeed not that compelling. The main result is mostly derived from applying Klochkov and Zhivotovskiy's techniques to a more restrictive (in the sense of problem well-behavedness constraints) and more general (in the sense of studying the "oracle" $A$) set-up. This might potentially bar the paper from entering the conference highlight.
>
>
> Answer: Thank you for recognizing our work. In public response, we highlight the novelty of our work, particularly focusing on both the novelty of the results and the techniques employed. Notably, our results are asymptotically optimal, which align with existing theories. We not only provide a more precise stability analysis tool under PL conditions, but we also offer new and improved insights into the relationship between algorithm stability and generalization in non-convex settings (Theorem 3). This has potential applications in non-convex scenarios as well.
>
>
>
>
> > Question 2: As mentioned in Questions, Can the authors briefly comment on the implications for 1-dimensional mean estimation (w/ empirical mean, M-estimators, etc.)? This might be a good illustrative example.
>
>
> Answer: In our first application, Empirical Risk Minimization (ERM) can be regarded as a specific type of M-estimation (Sen, 2018), where the objective function is defined based on the empirical risk derived from the training data. In this sense, ERM represents a particular case of M-estimation that focuses specifically on minimizing the empirical average of a loss function. When our objective function satisfies the assumptions outlined in Theorem 5 (namely, being Lipschitz continuous, strongly convexity and smoothness), we can substitute the corresponding parameters into Theorem 5 to arrive at our conclusion.
>
> Reference
>
> Bodhisattva Sen. A gentle introduction to empirical process theory and applications. Lecture Notes Section 1.2, Columbia University 11 (2018): 28-29.

---

> > ### Comment · Reviewer_9oQf · 2024-11-21
> >
> > Thank you for your detailed response. I would like to keep my scores unchanged.
> >
> > The authors might consider spelling out in full, in the 1-dimensional mean estimation case, the assumptions on the distribution and M-estimator and the corresponding convergence theorem, in the camera-ready version upon acceptance.
> >
> >
> >
> >
> > Best of luck,
> >
> > 9oQf

---

> > > ### Author Response · Authors · 2024-11-25
> > >
> > > Thank you for your valuable feedback. Here, we will provide a detailed example. If we denote the observed data as $X_1, X_2, \ldots, X_n$ drawn from an unknown distribution independently, and we assume that this distribution is bounded (for example, any random variable $X$ that follows this distribution satisfies $a \leq X \leq b$).
> > >
> > > To estimate the mean of the random variable, our objective function is defined as $f(\mu; X) = (X - \mu)^2$, which yields the usual least squares estimator. Using the definition of M-estimators, our goal becomes:
> > >
> > > $$
> > > \hat{\mu} = \arg\min_\mu \frac{1}{n} \sum_{i=1}^{n} (X_i - \mu)^2.
> > > $$
> > >
> > > We can easily verify that $f(\mu; X)$ satisfies the following properties: it is $4(b-a)$-Lipschitz, $2$-strongly convex, and $2$-smooth with respect to $\mu$. According to Theorem 5, we know that the mean estimated by this method, which minimizes the least squares loss (or the second central moment), will converge to
> > >
> > > $$
> > > \left( \frac{\mathbb{V}[X] \log{(1/\delta)}}{n} + \frac{\log^2 n \log^2(1/\delta)}{n^2} \right),
> > > $$
> > >
> > > as the sample size $n$ increases, where $\mathbb{V}[X]$ is the variance of the distribution. It’s worth noting that since this example pertains specifically to estimating the mean of a random variable, we have no additional parameters involved. Therefore, in this case, $F(w^*)$ in Theorem 5, which represents the variance of the distribution, is $O(1)$. Consequently, we cannot achieve a convergence rate of $1/n^2$. However, when our objective function employs more complex model functions, our method can achieve the faster rate of $O(1/n^2)$.
> > >
> > > If we haven't explained things clearly or if you have any questions, please feel free to reach out to us.

---

### Official Review · Reviewer_KqxP · 2024-11-03

**Soundness:** 3
**Presentation:** 3
**Contribution:** 2
**Rating:** 6
**Confidence:** 3

**Summary:**

Th authors of the paper obtain high probability excess risk bounds of order up to $\mathcal{O}(1/n^2)$ (up multiplicative logarithmic factors) for both empirical risk minimization problems and gradient descent algorithms. Towards this aim, the authors consider uniform stability in gradients instead of uniform stability of the loss function itself.

**Strengths:**

The paper is well-written and main contributions are clearly articulated. The fact that the bounds for ERM and SGD setting do not require lower bounding the sample size $n$ in terms of the problem dimension $d$ is also a significant improvement, for example, over, [Xu & Zeevi, 2024].

**Weaknesses:**

The technical novelty of the results is limited. Essentially the proof is based on an appropriate combination of [Zhang & Zhou, 2019] and [Klochkov & Zhivotovskiy, 2021]. Moreover, the excess risk of order $\mathcal{O}(1/n^2)$ can be obtained only in the setting $F(w*) = \mathcal{O}(1/n)$, that is, provided that the noise-at-the-optimum is rather small. The tail bound of order $\log^2(1/\delta)$ in Theorems $4$ - $6$ (in front of $1/n^2$ terms) also seems to be not optimal. For example, this $\log^2(1/\delta)$ regime does not appear in [Klochkov & Zhivotovskiy, 2021].

**Questions:**

Is it possible to improve scaling of the right-hand side in Theorems $4-6$ with $\log(1/\delta)$?

---

> ### Author Response · Authors · 2024-11-21
>
> Thank you for your review. Here are the replies for your questions.
>
> > Question 1: As mentioned in Weaknesses, the technical novelty of the results is limited. Essentially the proof is based on an appropriate combination of [Zhang \& Zhou, 2019] and [Klochkov \& Zhivotovskiy, 2021].
>
> Answer: We appreciate your feedback regarding the perceived novelty of our proofs and results. In public response, we have provided a detailed explanation of the novelty of our results, as well as the innovative aspects of our proof techniques. Notably, our results are asymptotically optimal, which align with existing theories. We not only provide a more precise stability analysis tool under PL conditions, but we also offer new and improved insights into the relationship between algorithm stability and generalization in non-convex settings (Theorem 3). This has potential applications in non-convex scenarios as well. We believe these innovations warrant consideration and appreciate your understanding.
>
> > Question 2: As mentioned in Weaknesses, moreover, the excess risk of order $O(1/n^2)$ can be obtained only in the setting $F(w^*) = O(1/n)$, that is, provided that the noise-at-the-optimum is rather small.
>
> Answer: Thank you for your valuable feedback. We appreciate your insights regarding the bounds we presented in our work. It is important to clarify that our upper bound of $O(1/n^2)$ is dependent on optimal parameters, but this does not imply poorer performance under alternative conditions.
> For instance, using PGD algorithm, the result in (Klochkov \& Zhivotovskiy, 2021) is
> $$
> \text{excess risk} = O\Big( \frac{\log(1/\delta)}{n} \Big).
> $$
> As a comparison, under the same assumptions (including the same iteration step $T = O(\log n)$), our result in Theorem 6 is
> $$
> \text{excess risk} = O\Big(\frac{\log^2(1/\delta)}{n^2} + \frac{ \log(1/\delta) F(w^*)}{n}\Big).
> $$
> As long as $ F(w^*) $ converges as $ n $ increases, our result outperforms better. Even when $F(w^*) = O(1)$, our PGD result remains comparable to those presented by (Klochkov \& Zhivotovskiy, 2021).
> Similarly, in the case of SGD, (Li \& Liu, 2021) demonstrated an upper bound of $O(1/n)$ using stability methods when $T = n^2$.
>  In contrast, our Theorem 8 yields an optimal upper bound
> $$
>  \text{excess risk}  = O \left(\frac{ \left\lceil \log_2 n \right\rceil^2 \log T \log^5(1/\delta)}{T}\right) + O \left(\frac{ \left\lceil \log_2 n \right\rceil^2 \log^2(1/\delta)}{n^2} + \frac{F(w^*) \log(1/\delta)}{n}\right).
>     $$
> According to our result, our bound is $O(1/n^2 + F(w^*)/n)$ under $T = n^2$. Should we aim for an $O(1/n)$ result, the assumptions can be relaxed to $F(w^*) = O(1)$ and $T=n$. This implies that, for the same learning rates, our method requires fewer iterations compared to existing stability analysis results. Furthermore, for a given number of iterations, our results provide a more granular analysis. Under certain additional assumptions, we can achieve an upper bound of the order $ O(1/n^2) $.
>
> > Question 3: As mentioned in Weaknesses, the tail bound of order $\log^2(1/\delta)$ in Theorems $4$-$6$ (in front of $1/n^2$ terms) also seems to be not optimal. For example, this $\log^2(1/\delta)$ regime does not appear in [Klochkov \& Zhivotovskiy, 2021].
>
> Answer: Indeed, in the work of (Klochkov & Zhivotovskiy, 2021), there is no appearance of the term $\log^2(1/\delta)$. Instead, their result includes $\log(1/\delta)$ as the coefficient in front of the $1/n$ term. Considering that for their bound to be meaningful, it must hold that $\log(1/\delta)/n < 1$. This implies that under the same conditions for meaningfulness, our result, which is $\log^2(1/\delta)/n^2$, is tighter.
>
>
> > Question 4: As mentioned in Questions, is it possible to improve scaling of the right-hand side in Theorems $4$-$6$ with $\log^2(1/\delta)$?
>
> Answer: The discussion regarding $\log^2(1/\delta)$ can be referenced in the answer to Question 3. Let me first explain the origin of the $\log^2(1/\delta)$ term. This term arises from the process of transforming moments into tail bounds. The approach used in this paper involves deriving tighter bounds on the gradients. Consequently, when this result is converted into a tail bound, it achieves a scale of $\log(1/\delta)/n$. By taking the PL condition into account and linking the excess risk to the gradient of the population risk, the final result incorporates an additional squared term.
>
> From this perspective, the presence of the $\log^2(1/\delta)$ term in our proof methodology is unavoidable. If one wishes to improve upon this term, it may require a different approach, considering perspectives other than simply refining the bounds on the gradients.
>
>
> Reference
>
> Yegor Klochkov, and Nikita Zhivotovskiy. Stability and deviation optimal risk bounds with convergence rate $O(1/n)$. NeurIPS 2021.
>
> Shaojie Li and Yong Liu. Improved learning rates for stochastic optimization: Two theoretical viewpoints. arXiv preprint arXiv:2107.08686, 2021.

---

> ### Author Response · Authors · 2024-11-25
> **Follow-up on Rebuttal Submission**
>
> Dear Reviewer KqxP,
>
> We hope this message finds you well. We are writing to follow up on the rebuttal we submitted several days ago. We wanted to check in to see whether it has addressed your concerns.
>
> If there are any remaining questions or if there are new issues you would like to discuss, please feel free to reach out. We appreciate your feedback and am eager to ensure that all aspects of the paper meet your expectations.
> Thank you for your time and consideration.
>
>
> Best regards,
>
> Authors

---

> > ### Comment · Reviewer_KqxP · 2024-11-29
> >
> > Dear authors,
> >
> > Thank you for your comments and response. I will keep my score and continue to support accepting the paper.
> >
> > Best,
> > KqxP.

---

> ### Author Response · Authors · 2024-12-04
>
> Thank you for your constructive feedback. We greatly appreciate your time and effort in reviewing our work. Your comments have been invaluable in helping us refine our manuscript.

---

### Author Response · Authors · 2024-11-21
**Public Response (Part I)**

Dear Reviewers,

Thank you very much for your thoughtful and thorough review of our manuscript. We sincerely appreciate the time you have dedicated to providing constructive feedback, which has been invaluable in refining our work.

In this response, we would like to emphasize our contributions and engage with the concerns raised regarding the novelty of both our results and our proof techniques.


# Novelty of Our Results


1. Our results are asymptotically optimal, which aligns with existing theories. For instance, in the case of ERM, according to the classical asymptotic theory, under some local regularity assumptions, when $ n \rightarrow \infty $, it is shown in the asymptotic statistics monographs (Vaart, 2000) that
$$
    \sqrt{n} (\hat{w}^*(S) - w^*) \stackrel{\rho}\longrightarrow N(0, H^{-1}Q H^{-1}),
$$
where $ \hat{w}^*(S) $ denotes the ERM algorithm, $ H = \nabla^2 F(w^*) $, $Q$ is the covariance matrix of the loss gradient at $w^*$ (also called Fisher's information matrix): $ Q = \mathbb{E} [\nabla f(w^*; z) \nabla f(w^*; z)^T ] $
($A^T$ denotes the transpose of a matrix $A$), and $\stackrel{\rho}\longrightarrow$ means convergence in distribution. The second-order Taylor expansion of the population risk around $w^*$ then allows to derive the same asymptotic law for the scaled excess risk $ 2n (F(\hat{w}^*(S)) - F(w^*)) $. Under suitable conditions, this asymptotic rate is usually theoretically optimal (Vaart, 1989). For example, when $f(w;z)$ is a negative log-likelihood, this asymptotic rate matches the Hajek-LeCam asymptotic minimax lower bound (Hajek, 1972; LeCam, 1972).
We then analysis the result in Theorem 5. In the proof of Theorem 4, before we use the self-bounded smoothness property $ || \nabla f(w^*;z) ||^2 \leq 4 \gamma f(w^*;z) $, we get the following result for Theorem 5
$$
    F(\hat{w}^*(S)) - F(w^*) \lesssim \frac{ \mathbb{E}[ ||\nabla f(w^*;z)||^2 ] \log(1/\delta)}{\mu n} + \frac{\log^2 (1/\delta)}{n^2}.
    $$
Our result is the finite sample version of the asymptotic rate given above, which characterizes the critical sample size sufficient to enter this ``asymptotic regime''. This is because the excess risk error $F(\hat{w}^*(S)) - F(w^*)$ can be approximated by the quadratic form $ (\hat{w}^*(S) - w^*)^T H (\hat{w}^*(S) - w^*) $. $ 1/\mu $ is a natural proxy for the inverse Hessian $H^{-1}$, and $\mathbb{E}[ ||\nabla f(w^*;z)||^2 ]$ is a natural proxy for Fisher's information matrix $Q$.
Furthermore, when discussing sample complexity, (Xu \& Zeevi, 2024) constructed a simple linear model to demonstrate the constant-level optimality of the sample complexity lower bound $\Omega(d \beta^2/\mu^2)$ under such conditions. Our theorem further reveals, through the use of stability methods, that this complexity lower bound can be independent of the dimensionality $d$.

---

> ### Author Response · Authors · 2024-11-21
> **Public Response (Part II)**
>
> 2. Currently, advancements in the relationship between stability and generalization are limited; most papers focus on utilizing stability tools to address application problems or to explore the stability of specific algorithms. In contrast, our analysis of stability bounds dependent on optimal parameters is an improvement in this area.
> Best result given by (Klochkov \& Zhivotovskiy, 2021, Theorem 1.1) is
> $$
>     F(A(S)) -F(w^{\ast})  \lesssim  (F_S(A(S)) - F_S(w^*(S))) + \mathbb{E}(F_S(A(S)) - F_S(w^*(S))) + (\beta \log n + 1/n)\log{(1/\delta)}.
> $$
> In constrast, our result in Theorem 4 is
> $$
>     F(A(S)) -F(w^{\ast}) \lesssim || \nabla F_S(A(S)) ||_2^2 + \frac{F(w^*) \log{(1/\delta)}}{n} + \frac{ \log^2(1/\delta)}{n^2} + \beta^2 \log^2 n \log^2 (1/\delta).
> $$
> We can see that when the algorithm optimization is sufficiently effective (i.e., $ F_S(A(S)) - F_S(w^*(S)) = 0, \nabla F_S(A(S)) = 0 $), our conclusions, despite requiring additional smoothness and PL conditions, provide a more granular analysis dependent on optimal parameters.
>
> For instance, when the algorithm's stability $ \beta = O(1/\sqrt{n}) $ the upper bound, according to (Klochkov \& Zhivotovskiy, 2021), can at most reach the order of $1/\sqrt{n}$ due to the algorithm's stability constraints. In contrast, our result shows that even under the assumption that $F(w^*) = O(1)$, treating $F(w^*)$ as a constant, we can achieve an order of $1/n$.
>
> Next, we discuss the fast rates of stability methods by considering a specific, more stable algorithm, denoted as $ \beta = O(1/n) $, our results simplify to $ \frac{F(w^*)}{n} + \frac{1}{n^2} $. In contrast, the results from (Klochkov \& Zhivotovskiy, 2021) yield $ \frac{1}{n} $ (omitting logarithmic terms). This indicates that existing results cannot surpass the $ \frac{1}{n} $ barrier, while our conclusions can tighten as the condition on $ F(w^*) $ becomes stronger. For example, when $ F(w^*) = O(\frac{1}{\sqrt{n}}) $, our results can achieve a rate of $ O(\frac{1}{n^{3/2}}) $. In other words, as long as $ F(w^*) $ converges as $ n $ increases, our results outperform existing ones. Even in cases where $ F(w^*) = O(1) $, which is a constant independent of $ n $, our results are comparable to those presented by (Klochkov \& Zhivotovskiy, 2021).
>
> On the other hand, while it may seem that we have introduced additional smoothness and PL conditions, extensive research on specific algorithms shows that, to maintain good stability with high probability, smoothness and strong convexity conditions are almost essential—for instance, (Klochkov \& Zhivotovskiy, 2021) for PGD. Essentially, we are effectively utilizing the smoothness and PL conditions to explore the relationship between stability and generalization error. Thus, when addressing the ultimate problem, we achieve better results without adding extra conditions.
>
> Finally, this approach can be seamlessly extended to various areas, such as differentially private models, pairwise learning, minimax problems et al., leading to more refined, parameter-dependent stability bounds across a wider array of algorithms.
>
>
>
> 3. Our methodology has the potential for extensive applications.  For application, we have achieved the best current results for empirical risk minimization (ERM), projected gradient descent (PGD), and stochastic gradient descent (SGD). It is important to clarify that our upper bound of $ O(1/n^2) $ is based on optimal parameters; however, this does not imply poorer performance under alternative conditions. As discussed in the second point, the original method requires the algorithm's stability to be on the order of $ 1/n $ in order to achieve an upper bound of $ 1/n $. In contrast, our approach only necessitates that the specific algorithm's stability is at the order of $ 1/\sqrt{n} $ to reach the same $ 1/n $ upper bound. This means our method is less dependent on assumptions in practical applications.
>
>
>
>
> 4. We also see potential applications of our approach in non-convex settings. While we focused primarily on maintaining the readability of our manuscript, which limits the discussion about non-convex settings in Remark 4. We admit that exploring high-probability stability for specific algorithms in non-convex setting is indeed challenging. However, our core result (Theorem 3) does not rely on convexity assumptions, making it an invaluable tool for discussing algorithm stability and generalization error bounds in non-convex settings.

---

> ### Author Response · Authors · 2024-11-21
> **Public Response (Part III)**
>
> # Novelty in Proof Techniques
>
> 1. For Theorem 1, in our writing, we discovered that the constant in Marcinkiewicz-Zygmund's inequality for random variables taking values in a Hilbert space could be further improved. Consequently, we included this part in the paper. Considering that Theorem 1 has other broad applications, we report this result as well. Although it is the constant-level improvement. The proof was challenging, as it involved establishing the best constant in Marcinkiewicz-Zygmund's inequality for random variables taking values in a Hilbert space, which has its foundations in Khintchine-Kahane's inequality. To prove the best constant, we utilized Stirling's formula for the Gamma function to construct appropriate functions, establishing both upper and lower bounds. Then using Mean Value Theorem, this approach ultimately led to the convergence of the constant as $p$ approaches infinity.
>
> 2. For Theorem 3, although this theorem is inspired by (Klochkov \& Zhivotovskiy, 2021), it is essential to note that prior work predominantly focuses on one-dimensional random variables. Our approach, however, confronted additional complexities due to the involvement of random vectors and their norms. In one-dimensional cases, the condition of a variable value being $0$ is equivalent to its absolute value being $0$. However, the Euclidean norm of a vector being $0$ does not imply that the vector itself is the $0$ vector, adding significant complexity to our proof. This distinction necessitated a more nuanced analysis during the proofs.
>
> 3. For Theorem 4, the proof for this theorem primarily exploits the self-bounded properties of functions, facing challenges akin to those highlighted in Theorem 3, related to the interaction between norms $ || \cdot || $ and expectation symbols $ \mathbb{E} $. While we can sometimes apply the Jensen's inequality, there are instances where we must innovatively navigate around such relationships within the function construction. This complexity does not arise for scalar variables, and we think that this provide a pathway for future inquiries into the stability of algorithms involving vector norm considerations.
>
> We hope that our detailed response helps clarify the novelty and significance of our manuscript. We are grateful for your insights and look forward to your feedback.
>
>
>
> Authors
>
>
>
> Reference
>
>
> Aad W Van der Vaart. Asymptotic statistics, volume 3. Cambridge university press, 2000.
>
> And W Van der Vaart. On the asymptotic information bound, The Annals of Statistics (1989) 1487-1500.
>
> Jaroslav Hajek. Local asymptotic minimax and admissibility in estimation. Berkeley symposium on mathematical statistics and probability, Volume 1: Theory of Statistics. 1972.
>
> L Le Cam et al. Limits of experiments. Berkeley symposium on mathematical statistics and probability, Volume 1: Theory of Statistics. 1972.
>
> Yunbei Xu and Assaf Zeevi. Towards optimal problem dependent generalization error bounds in statastical learning theory. Mathematics of Operations Research 2024.
>
> Yegor Klochkov, and Nikita Zhivotovskiy. Stability and deviation optimal risk bounds with convergence Rate $O(1/n)$. NeurIPS 2021.

---

### Meta-Review · Area_Chair_idUY · 2024-12-22

**Metareview:**

This paper derives high-probability generalization bounds based on stability. The main contribution of this work is a risk bound of order O(1/n^2) under certain conditions. Further, authors discuss implications to ERM problems, gradient descent and SGD.

This paper was reviewed by four reviewers and received the following Scores/Confidence: 5/5, 5/3, 6/3, 8/3. I think paper is studying an interesting topic, and the results are intriguing but authors are not able to convince half the reviewers sufficiently well about the technical novelty. The following concerns were brought up by the reviewers:

- Results in this paper can be derived by a straightforward combination of existing work. This limits the technical contribution/novelty significantly given that this is the main contribution.
- Dependence on the PL constant is not good. The overall rate depends quadratically on this parameter, which can be extremely small in practice.
- Conditions and their practical implications are not sufficiently discussed in the paper.


Three reviewers are not particularly excited about the paper. As such, based on the reviewers' suggestion, as well as my own assessment of the paper, I recommend not including this paper to the ICLR 2025 program.

**Additional Comments On Reviewer Discussion:**

Authors rebuttal addressed a subset of reviewers' concerns. However, the main concern about the technical novelty remains.

---

### Decision · Program_Chairs · 2025-01-22

Reject